# MANIFOLD PRESERVING GUIDED DIFFUSION

**Yutong He**[1,2,*,†]**, Naoki Murata**[2,*] **, Chieh-Hsin Lai**[2] **, Yuhta Takida**[2]
**Toshimitsu Uesaka**[2] **, Dongjun Kim**[2,†] **, Wei-Hsiang Liao**[2] **, Yuki Mitsufuji**[2,3]
**J. Zico Kolter**[1] **, Ruslan Salakhutdinov**[1] **, Stefano Ermon**[4]
[1]Carnegie Mellon University , [2]Sony AI , [3]Sony Group Corporation , [4]Stanford University
yutonghe@cs.cmu.edu, naoki.murata@sony.com

## ABSTRACT

Despite the recent advancements, conditional image generation still faces challenges of cost, generalizability, and the need for task-specific training. In this paper, we propose **M**anifold **P**reserving **G**uided **D**iffusion (**MPGD**), a *training-free* conditional generation framework that leverages pretrained diffusion models and off-the-shelf neural networks with minimal additional inference cost for a broad range of tasks. Specifically, we leverage the manifold hypothesis to refine the guided diffusion steps and introduce a shortcut algorithm in the process. We then propose two methods for on-manifold training-free guidance using pre-trained autoencoders and demonstrate that our shortcut inherently preserves the manifolds when applied to latent diffusion models. Our experiments show that MPGD is efficient and effective for solving a variety of conditional generation applications in low-compute settings, and can consistently offer up to $3.8\times$ speed-ups with the same number of diffusion steps while maintaining high sample quality compared to the baselines. Our code is available via the project page here.

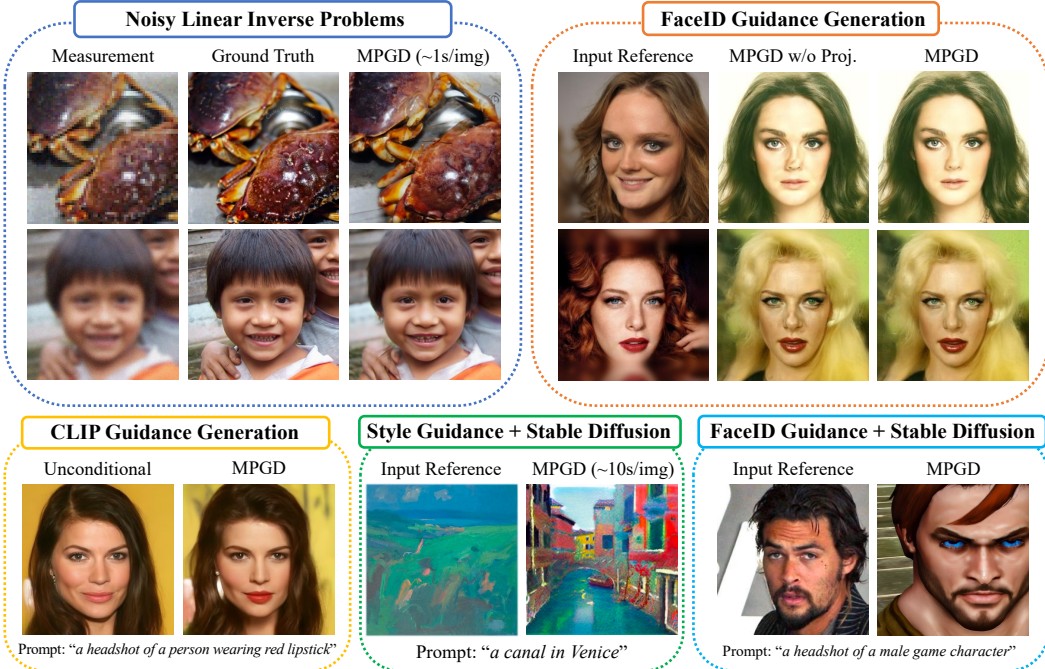

Figure 1: Our proposed **MPGD** as a training-free sampling method for both pre-trained pixel-space diffusion models and latent diffusion models in a variety of conditional generation applications. MPGD can be applied to a broad range of tasks with minimal sampling time and high sample quality.

---

[*]Equal contribution
[†]Work done during an internship at Sony AI

# 1 INTRODUCTION

Generative modeling has witnessed extraordinary breakthroughs in recent years (OpenAI, 2023; Ho et al., 2020; Rombach et al., 2021). Conditional generation, in particular, stands out as a crucial task, as it underlies solutions to several real-world problems, including image restoration, super-resolution, and creation of content with specific styles. However, despite the significant attention, conditional generation still faces its own set of challenges related to cost and generalizability: typical conditional generation requires either additional task-specific training, data collection, model architecture designs, or extra assumptions about the conditional generation tasks (Zhang et al., 2023; Ruiz et al., 2023; Isola et al., 2017; Park et al., 2019; Li et al., 2023). These requirements not only escalate costs but also restrict the range of applications and potential users.

Recent developments in diffusion models offer potential solutions to overcome these challenges (Song et al., 2021b; Dhariwal & Nichol, 2021; Wallace et al., 2023). In particular, Chung et al. (2023a) and many of its followup works (Song et al., 2023b; Bansal et al., 2023; Yu et al., 2023) use off-the-shelf loss functions to guide the sampling process. While these methods avoid extra training of diffusion models, their reliability remains inconsistent – at times they exhibit impressive performance, while in other instances they struggle to produce realistic images. Moreover, these methods tend to be extremely slow because they rely on extensive sampling time optimization and/or exceedingly large number of diffusion time steps to produce satisfactory samples. Above all, current literature has very limited understanding of when, why, and how these methods succeed or fail, making it difficult to design practical implementations in real-life applications.

In this paper, we propose **M**anifold **P**reserving **G**uided **D**iffusion (**MPGD**), a framework for conditional generation using unconditionally pretrained diffusion models with (1) no extra training (2) minimal additional computation and sampling time (3) generalizability to a broad range of tasks and (4) high sample quality. Central to our method, we leverage the so-called manifold hypothesis – the fact that the real data does not lie within the totality of the pixel space, but instead lies on a very small underlying manifold. Our key idea is that instead of guiding the diffusion process without constraint (until the last time step, when it hopefully arrives at the manifold), we can project the guidance to the manifold, via its tangent spaces, throughout the diffusion process. Moreover, when using the DDIM (Song et al., 2021a) sampling approach, we also show the method leads to an efficient "shortcut" for guidance gradients that saves both time and memory and substantially improves the sample quality over competing approaches in low-resource settings.

With this new framework, we derive several novel methods to perform training-free guided diffusion generation. We specifically analyze two different practical approaches to manifold projection using off-the-shelf unconditionally pretrained autoencoders for pixel-space diffusion models. We also show that applying our shortcut to latent diffusion models is naturally manifold preserving and can significantly improve the sample quality and the inference speed. Finally, we can extend the current framework to incorporate multi-step optimization algorithms to further improve the performance.

We empirically test our methods against competitive training-free guided diffusion baselines on various conditional generation tasks, including solving noisy linear inverse problems, human face generation with facial recognition model (FaceID) guidance, and text-to-image generation guided by a certain input style, as illustrated in Figure 1. Experiments show that our methods find a better tradeoff between fidelity and controllability compared to the baseline methods and are able to consistently achieve up to $3.8\times$ speed-ups while maintaining high sample quality.

# 2 CONVENTIONAL TRAINING FREE GUIDED DIFFUSION

## 2.1 PROBLEM FORMULATION

Let $x \in \mathcal{X} \subset \mathbb{R}^d$ be a $d$-dimensional sample in the support $\mathcal{X}$ of the data distribution and $y \in \mathcal{Y}$ be the given input condition such as a text description and an input reference image. In this paper, we aim at solving the problem of conditional generation by attempting to sample from the posterior distribution $p(x|y)$. We assume we have access to pretrained generative models for the prior distribution $p(x)$, and a differentiable loss function $L(x; y)$ giving us the posterior $p(x|y) \propto p(x)\exp(-L(x; y))$.

We target solutions that are: (1) **Training free:** Pretrained models can be deployed without extra training, (2) **Low cost:** The method should require minimal additional computational resources and time, (3) **Generalizable:** We only require black-box access to the loss function and its gradients, (4) **High quality:** The samples should come from a distribution that is close to the true posterior.

## 2.2 DIFFUSION MODELS

The score-based generative models (Song & Ermon, 2019; Song et al., 2021b), or diffusion models (Sohl-Dickstein et al., 2015; Ho et al., 2020), enable sampling from a clean data distribution by iteratively using the time-dependent score function $\nabla_{x_t} \log p_t(x_t)$ for noisy data $x_t$, where $t \in [0, T]$ and $T > 0$. In DDPM (Ho et al., 2020), noisy data $x_t$ is obtained by adding Gaussian noise $\epsilon_t \sim \mathcal{N}(0, I)$ to the scaled clean data $x \sim p(X)$, as $x_t = \sqrt{\bar{\alpha}_t} x + \sqrt{1 - \bar{\alpha}_t} \epsilon_t$ where $\bar{\alpha}_t > 0$ is a scaling parameter. The score function is frequently parameterized through a denoiser $\epsilon_\theta(x_t, t)$ trained with the loss function $\mathbb{E}_{t,x,\epsilon_t}[\|\epsilon_t - \epsilon_\theta(x_t, t)\|_2]$, so that $\epsilon_\theta$ estimates Gasussian noise included in noisy sample $x_t$. In the inference time, we can obtain clean data samples by applying the score function iteratively to noisy samples (Song & Ermon, 2019). In particular, DDIM (Song et al., 2021a) performs each step of the sampling with the update rule

$$x_{t-1} = \sqrt{\bar{\alpha}_{t-1}} \left( \frac{x_t - \sqrt{1 - \bar{\alpha}_t} \epsilon_\theta(x_t, t)}{\sqrt{\bar{\alpha}_t}} \right) + \sqrt{1 - \bar{\alpha}_{t-1} - \sigma_t^2} \epsilon_\theta(x_t, t) + \sigma_t \epsilon_t, \quad (1)$$

where the first term on the right-hand side corresponds to a direct estimation of the clean data $x$ from the noisy data $x_t$ by the diffusion model, which is derived from Tweedie's formula (Efron, 2011), denoted as $x_{0|t}$. In this formulation, $\sigma_t = \sqrt{(1 - \bar{\alpha}_{t-1})/(1 - \bar{\alpha}_t)} \sqrt{1 - \bar{\alpha}_t / \bar{\alpha}_{t-1}}$ corresponds to the DDPM sampling, and with $\sigma_t = 0$ the sampling procedure becomes deterministic.

## 2.3 RELATED WORKS ON TRAINING-FREE GUIDED DIFFUSION

Building upon early efforts such as Song et al. (2021b) and Meng et al. (2022) (detailed discussion is in the Appendix A), many recent papers, including classifier-guidance diffusion (Song et al., 2021b; Dhariwal & Nichol, 2021), DPS (Chung et al., 2023a), ΠGDM (Song et al., 2022), Free-DoM (Yu et al., 2023), UGD (Bansal et al., 2023), attempt to leverage pretrained diffusion models for various conditional generation tasks. The common underlying concept shared by these methods is to decompose a conditional score function $\nabla_{x_t} \log p(x_t|y)$ into the unconditional score function and the loss-based term $\nabla_{x_t} \log p(x_t|y) = \nabla_{x_t} \log p(x_t) + \nabla_{x_t} L_t(x_t; y)$.

The discretized sampling procedure with the decomposed conditional score can be interpreted as a two-step process based on the additivity of the terms. When an initial noisy sample $x_t$ is given, a denoised $x_{t-1}$ is obtained by the sampling process with the unconditional diffusion model. Subsequently, $x_{t-1}$ is further updated using the gradient of $L_t(x_t; y)$ with respect to $x_t$.

In particular, if we assume the noisy log likelihood can be accessed by a time-dependent differentiable function, as $L_t(x_t; y)$, the second step can be regarded as an optimization step by gradient descent to minimize the guidance loss in the vicinity of the denoised sample $x_t$:

$$x_t \leftarrow x_t - \rho_t \nabla_{x_t} L_t(x_t; y), \quad (2)$$

where $\rho_t$ is a time-dependent step size parameter. Hence, the optimization problem solved in this step is represented as:

$$\min_{x_t' \in N(x_t) \subset \mathbb{R}^d} L_t(x_t'; y), \quad (3)$$

where $N(x_t) = \{x \in \mathbb{R}^d \mid d(x, x_t) < r_t\} \subset \mathbb{R}^d$ is a neighbourhood around $x_t$ in $\mathbb{R}^d$ bounded by some radius $r_t$ which is related to the optimization step size $\rho_t$.

However, one caveat exists: usually the pretrained guidance loss function is only defined on clean data $x$ instead of noisy data $x_t$. In other words, we usually only have access to an $L$ that is trained on clean data rather than $L_t$'s that are trained on noisy data. To solve this problem, Chung et al. (2023a) uses the clean data estimation $x_{0|t} = \frac{1}{\sqrt{\bar{\alpha}_t}}(x_t - \sqrt{1 - \bar{\alpha}_t} \epsilon_\theta(x_t, t))$ from the Tweedie's formula (Efron, 2011) as a point estimation of the true loss term. Therefore, we can rewrite the update rule as

$$x_t \leftarrow x_t - \rho_t \nabla_{x_t} L(x_{0|t}; y), \quad (4)$$

and many previous methods follow this formulation. For example, DPS deals with inverse problems of the form $y = \mathcal{A}(x) + z$, where $\mathcal{A}(\cdot)$ is a differentiable function of $x$ and $z$ is an additive observation

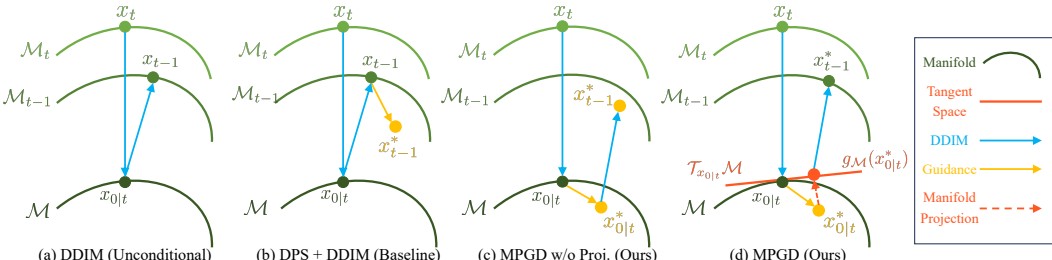

Figure 2: A schematic overview of our proposed approaches and an illustrative comparison with DDIM (Song et al., 2021a) and DPS (Chung et al., 2023a).

noise. In cases with Gaussian observation noise, it defines the loss as $L(x; y) = \|y - \mathcal{A}(x)\|_2^2$. LGD (Song et al., 2023b), UGD (Bansal et al., 2023), and FreeDoM (Yu et al., 2023) offer more flexibility in designing loss functions, allowing them to handle a variety of tasks. For instance, in the case of FaceID-guided generation, FreeDoM adopts a loss that calculates the $\ell_2$ distance between the features obtained by a facial recognition model and the features for the target face image.

## 3 ISSUES IN THE PREVIOUS FORMULATION: THE MANIFOLD HYPOTHESIS

In the previous formulation in equation 3, the neighborhood $N(x_t)$ for the optimization objective resides the ambient space $\mathbb{R}^d$. However, in practice the data lies in a much lower-dimensional space than the ambient space. Typically, the following assumptions can be made:

**Assumption 1** (Manifold Hypothesis). *The support $\mathcal{X}$ of the data distribution of interest lies on a $k$ dimensional manifold $\mathcal{M}$ that is embedded in a $d$ dimensional ambient space $\mathbb{R}^d$ such that $k \ll d$.*

**Assumption 1.1** (Linear Subspace Manifold Hypothesis). *The data manifold $\mathcal{M} \subset \mathbb{R}^d$ is a linear subspace of dimension $k \ll d$.*

Chung et al. (2022; 2023b) have shown that with linear subspace manifold hypothesis and Gaussian annulus theorem (Blum et al., 2020), at a certain diffusion time step $t$, the noisy data $x_t$ also probabilistically concentrate on a manifold $\mathcal{M}_t$ that has dimension $d-1$ and a shell-like geometric structure to the original data manifold $\mathcal{M}$. Here we provide an extended version of the proposition stated mathematically as follows:

**Proposition 1** (Concentration of Noisy Samples (Informal, extended from Chung et al. (2022; 2023b))). *Define $d(x, \nu, \mathcal{M}) := \inf_{x' \in \mathcal{M}} \|x - \nu x'\|_2$ for $\nu > 0$, and $B(\mathcal{M}; r) := \{x \in \mathbb{R}^d \mid d(x, 1, \mathcal{M}) < r\}$ for $r > 0$. Consider the distribution of noisy data $p_t(x_t) := \int p(x_t|x)p(x)dx$, where $p(x_t|x) := \mathcal{N}(\sqrt{\bar{\alpha}_t}x, (1-\bar{\alpha}_t)I)$. Then under Assumption 1.1, $p_t(x_t)$ is "probabilistically concentrated" on the $(d-1)$-dimensional manifold $\mathcal{M}_t$ defined as*

$$\mathcal{M}_t := \{x \in \mathbb{R}^d \mid d(x, \sqrt{\bar{\alpha}_t}, \mathcal{M}) = \sqrt{(1-\bar{\alpha}_t)(d-k)}\}.$$

The formal version of Proposition 1 and its proof is provided in Appendix B.

As a result, because the neighborhood resides in the ambient space rather than on the manifold $\mathcal{M}_t$, not all points in $N(x_t)$ in Equation 3 are necessarily close to or included in the manifold $\mathcal{M}_t$. This finding, which is empirically verified in Figure 3, suggests that the results obtained through optimization within $N(x_t)$ may deviate from $\mathcal{M}_t$ and adversely affect the evaluation of the score function (or sampling by the diffusion model) in the following steps, since the score function is trained only with samples close to $\mathcal{M}_t$ due to Proposition 1. Therefore, optimization within $N(x_t)$ cannot guarantee that the final result will correspond to a realistic image. In practice, we observe that methods such as DPS, UGD and FreeDoM require detailed fine-tuning of step size scheduling, techniques such as repainting (Lugmayr et al., 2022), or a large number of diffusion time steps to ensure that gradient updates don't deteriorate the final results.

## 4 MANIFOLD PRESERVING TRAINING-FREE GUIDED DIFFUSION

Based on our analysis in Section 3, we reformulate the objectives in Section 2.3 to address the manifold hypothesis and propose the following framework to perform on-manifold guided diffusion.

### 4.1 OBJECTIVE

We first rewrite the minimization objective in Equation 3 by considering a different neighborhood than $N(x_t)$. Since $\mathcal{M}_t$ is a manifold, the "neighborhood" can be represented as an open subset of the tangent space $\mathcal{T}_{x_t}\mathcal{M}_t$ of $x_t$, which is homeomorphic to an open subset in $\mathbb{R}^k$ for $k \ll d$ (Shao et al., 2018). Intuitively, optimizing on a small neighborhood on the tangent spaces allows us to only make "reasonable" changes to the samples. With tangent spaces, we can write our objective as

$$\min_{x'_t \in N_{\mathcal{T}}(x_t) \subset \mathcal{T}_{x_t}\mathcal{M}_t} L(\frac{1}{\sqrt{\bar{\alpha}_t}}(x'_t - \sqrt{1-\bar{\alpha}_t}\epsilon_\theta(x'_t, t)); y), \tag{5}$$

where $N_{\mathcal{T}}(x_t) = \{x \in \mathcal{T}_{x_t}\mathcal{M}_t \mid d(x, x_t) < r_t\} \subset \mathcal{T}_{x_t}\mathcal{M}_t$ is a small neighborhood around $x_t$ in its the tangent space $\mathcal{T}_{x_t}\mathcal{M}_t$ and $r_t$ is the radius of the neighborhood related to the optimization step size $\rho_t$. The objective is to find the point $x_t^*$ in that neighborhood such that its Tweedie's estimation $x_{0|t}^* = \frac{1}{\sqrt{\bar{\alpha}_t}}(x_t^* - \sqrt{1-\bar{\alpha}_t}\epsilon_\theta(x_t^*, t))$ of the clean data best aligns with the given conditions $y$.

### 4.2 METHOD

Conventionally we can estimate the tangent spaces of a data manifold using an autoencoder (Shao et al., 2018; Bordt et al., 2023; Srinivas et al., 2023; Anders et al., 2020). The key idea is that, the information bottleneck in autoencoders naturally incorporates manifold hypothesis and a well-trained autoencoder yields latent representations that implicitly capture the local lower dimensional coordinates for the data manifold. However, while most off-the-shelf autoencoders are trained on the clean data, notice that in Equation 5 we need access to the tangent spaces of the noisy samples $x_t$ in $\mathcal{M}_t$. So how should we achieve this goal with only access to clean data manifold $\mathcal{M}$?

#### 4.2.1 THE MPGD SHORTCUT

Combining the results of Proposition 1 and Lemma 2 in Appendix B, we first obtain the following theorem that facilitates us to perform manifold preserving guidance with access to only the clean data manifold: If a guidance gradient preserves the manifold for clean data, it also brings a noisy sample on a noisy manifold.

**Theorem 1.** *(Informal) Assume the gradient $\nabla_{x_{0|t}}L(x_{0|t}; y)$ lies on the tangent space $\mathcal{T}_{x_{0|t}}\mathcal{M}$, and the diffusion model $\epsilon_\theta(x_t, t)$ is optimal. Then with Assumption 1.1, scalar $c_t > 0$ and update rule*

$$x_{t-1} = \sqrt{\bar{\alpha}_{t-1}}(x_{0|t} - c_t\nabla_{x_{0|t}}L(x_{0|t}; y)) + \sqrt{1-\bar{\alpha}_{t-1}-\sigma_t^2}\epsilon_\theta(x_t, t) + \sigma_t\epsilon_t, \tag{6}$$

*we can obtain an $x_{t-1}$ whose marginal distribution is probabilistically concentrated on $\mathcal{M}_{t-1}$.*

The formal statement, proof and discussions are provided in the appendix. Therefore, we can derive a simplified update rule for manifold preserving guided diffusion that only requires access to the tangent spaces of the clean data manifold $\mathcal{M}$ as long as we can ensure that $\nabla_{x_{0|t}}L(x_{0|t}; y))$ is also on the tangent space $\mathcal{T}_{x_{0|t}}\mathcal{M}$:

$$x_{0|t} \leftarrow x_{0|t} - c_t\nabla_{x_{0|t}}L(x_{0|t}; y) \qquad \text{(a step of gradient descent)} \tag{7}$$

$$x_{t-1} \leftarrow \sqrt{\bar{\alpha}_{t-1}}x_{0|t} + \sqrt{1-\bar{\alpha}_{t-1}}\epsilon_\theta(x_t, t) \qquad \text{(rescaling of the clean data and the noise)} \tag{8}$$

This algorithm can also be intuitively viewed as updating the DDIM clean data estimation $x_{0|t}$ at time $t$ with the guidance gradient with respect to that estimation. While this approach is generally faster since it doesn't require computing the gradient with respect to $x_t$ for the score function, we should note that it requires the guidance gradient $\nabla_{x_{0|t}}L(x_{0|t}; y))$ to reside in the tangent space of the manifold $\mathcal{T}_{x_{0|t}}\mathcal{M}$, leading to on-manifold samples. Therefore, we further investigate the ways to project the guidance onto the manifold, and refer to this shortcut as manifold preserving guided diffusion without projection, **MPGD w/o Proj.**.

#### 4.2.2 MANIFOLD PROJECTION WITH (PERFECT) AUTOENCODERS

Now that we have established an algorithm that only requires access to the clean data manifold, we can use an off-the-shelf autoencoder to project the guidance onto the tangent spaces. To demonstrate the process, we first showcase the derivation where we have access to a perfect autoencoder. Note that the following is inspired by (Shao et al., 2018; Anders et al., 2020), but does not perfectly match with them.

**Algorithm 1** MPGD for pixel diffusion models

1: $x_T \sim \mathcal{N}(0, I)$
2: **for** $t = T, \ldots, 1$ **do**
3:     $\epsilon_t \sim \mathcal{N}(0, I)$
4:     $x_{0|t} = \frac{1}{\sqrt{\bar{\alpha}_t}}(x_t - \sqrt{1 - \bar{\alpha}_t}\epsilon_\theta(x_t, t))$
5:     **if** requires manifold projection **then**
6:         $x_{0|t} = g_{\mathcal{M}}(x_{0|t}, L(x_{0|t}; y), c_t)$
7:     **else**
8:         $x_{0|t} = x_{0|t} - c_t \nabla_{x_{0|t}} L(x_{0|t}; y)$
9:     **end if**
10:    $x_{t-1} = \sqrt{\bar{\alpha}_{t-1}} x_{0|t}$
11:        $+ \sqrt{1 - \bar{\alpha}_{t-1} - \sigma_t^2}\epsilon_\theta(x_t, t) + \sigma_t \epsilon_t$
12: **end for**
13: **return** $x_0$

**Algorithm 2** MPGD for latent diffusion models

1: $z_T \sim \mathcal{N}(0, I)$
2: **for** $t = T, \ldots, 1$ **do**
3:     $\epsilon_t \sim \mathcal{N}(0, I)$
4:     $z_{0|t} = \frac{1}{\sqrt{\bar{\alpha}_t}}(z_t - \sqrt{1 - \bar{\alpha}_t}\epsilon_\theta(z_t, t))$
5:     $z_{0|t} = z_{0|t} - c_t \nabla_{z_{0|t}} L((D(z_{0|t}); y)$
6:     $z_{t-1} = \sqrt{\bar{\alpha}_{t-1}} z_{0|t}$
7:        $+ \sqrt{1 - \bar{\alpha}_{t-1} - \sigma_t^2}\epsilon_\theta(z_t, t) + \sigma_t \epsilon_t$
8: **end for**
9: **return** $x_0 = D(z_0)$

**Algorithm 3** $g_{\mathcal{M}}$: On-Manifold Guidance

1: **if** MPGD-AE **then**
2:     $x_{0|t} = x_{0|t} - c_t \nabla_{x_{0|t}} L(D(E(x_{0|t})); y)$
3: **else if** MPGD-Z **then**
4:     $z_{0|t} = E(x_{0|t})$
5:     $z_{0|t} = z_{0|t} - c_t \nabla_{z_{0|t}} L(D(z_{0|t}); y)$
6:     $x_{0|t} = D(z_{0|t})$
7: **end if**
8: **return** $x_{0|t}$

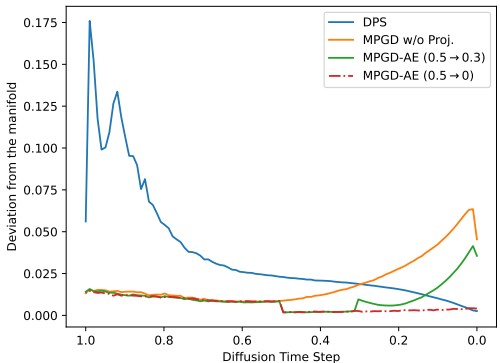

Figure 3: We analyze the deviation from the manifold throughout the diffusion process for different methods (details are in Appendix C).

**Assumption 2.** *(Perfect Autoencoder) Assume that for the support $\mathcal{X} \subset \mathcal{M}$ of the data distribution, there exists a perfect autoencoder with encoder $E : \mathcal{X} \to \mathcal{Z}$ and decoder $D : \mathcal{Z} \to \mathcal{X}$ with $\mathcal{Z} = \mathbb{R}^k$ for $k < d$. This autoencoder exhibits zero reconstruction error for each point on $x_0 \in \mathcal{M}$, i.e., $x_0 = D \circ E(x_0)$. Furthermore, the decoder $D$ is surjective to $\mathcal{M}$, and the encoder function $E$ and the decoder function $D$ form a pseudoinverse pair (Sorrenson et al., 2023), implying $E \circ D$ is an identity map.*

Under the assumptions of a perfect autoencoder, we can obtain gradients that preserve the manifold, as supported by the following theorem, the proof of which is provided in the appendix.

**Theorem 2.** *If an autoencoder with encoder $E$ and decoder $D$ is a perfect autoencoder for the support $\mathcal{X} \subset \mathcal{M}$ of the data distribution, then $\nabla_{x_0} L(D(E(x_0)); y) = \frac{\partial L}{\partial D} \frac{\partial D}{\partial E} \frac{\partial E}{\partial x_{0|t}} \in \mathcal{T}_{x_0} \mathcal{M}$.*

Therefore, to achieve the local minima of the objective function in Equation 5, we can modify the update rules in Equation 7 as:

$$x_{0|t} \leftarrow x_{0|t} - c_t \nabla_{x_{0|t}} L(D(E(x_{0|t})); y) \tag{9}$$

where with linear manifold hypothesis, the guided $x_{0|t}$ is on the tangent space $\mathcal{T}_{x_{0|t}} \mathcal{M}$ and $x_{t-1}$ is concentrated on $\mathcal{M}_{t-1}$. We refer to this method as **MPGD-AE**.

Although our analysis consists of perfect autoencoder assumption, in practice, we find that well-trained imperfect autoencoders such as VQGAN's (Esser et al., 2020) also have similar effects for mapping the guidance to the data manifold. In Figure 3, we empirically verifies VQGAN's manifold preserving ability by using it as the manifold projection function of MPGD-AE. Detailed discussion is included in Appendix C. We also present further analysis and experiments with empirically well-trained imperfect autoencoders in Section 5.

**Manipulating the Latents** The idea of preserving the manifold using a perfect autoencoder can be also achieved as the following: Rather than updating $x_{0|t}$ with gradient decent, we modify the encoded latent variable $z_{0|t}$ with its gradients instead. After updating $z_{0|t}$, we then map it back to the data space with the decoder to obtain a new estimation of $x_{0|t}$. We refer to this method as **MPGD-Z**. The details and comparison among three methods are provided in Algorithms 1, 2, and 3, where we refer to the manifold projection function as $g_{\mathcal{M}}$. We also provide additional analysis in Appendix B.4.

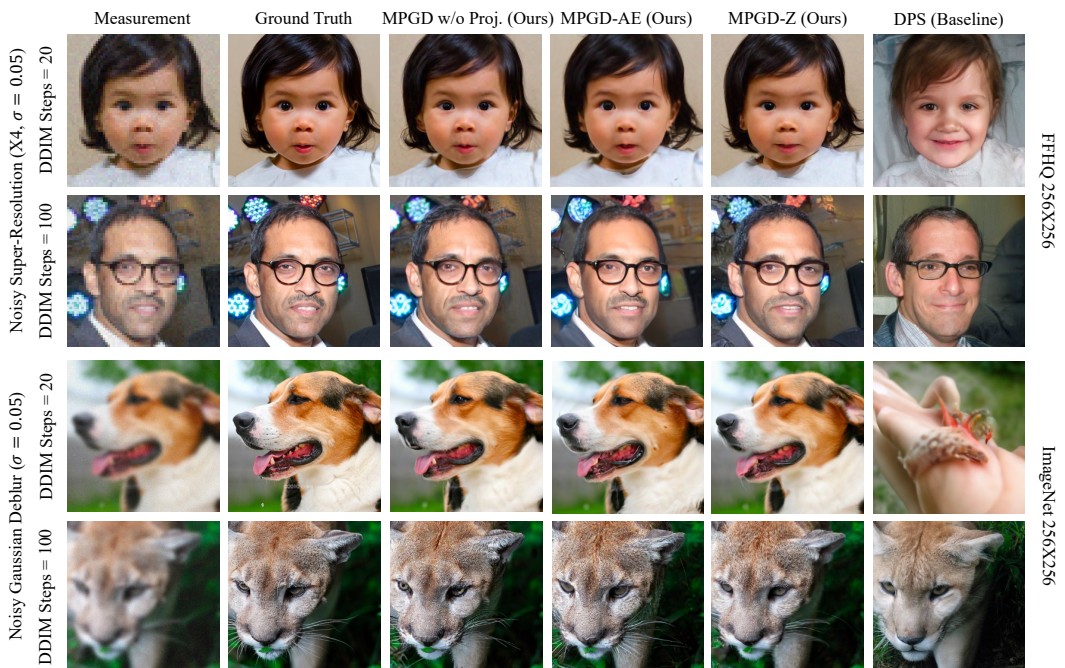

Figure 4: Qualitative examples of solving noisy linear inverse problems with our proposed MPGD and baseline DPS.

### 4.2.3 MPGD WITH LATENT DIFFUSION MODELS

Latent diffusion models (LDM), proposed by Rombach et al. (2021), is a procedure to gradually transform a sample $z_T \in \mathbb{R}^k$ to $z_0 \in \mathcal{Z}$ where $\mathcal{Z}$ is the same space as the latent space of a well-trained autoencoder such as a VQGAN (Esser et al., 2020) or VQVAE (van den Oord et al., 2017).

With the same intuition, we can also manipulate the latents in LDM using the same technique described in the previous section. Since LDM operates on the latent space of the autoencoder, the decoded latent guidance $D(\nabla_{z_{0|t}} L(D(z_{0|t}); y))$ is on the tangent spaces of the data manifold. Therefore, with linear manifold hypothesis, the final sample $x_0$ is on the manifold $\mathcal{M}$. We refer to this approach with LDM as **MPGD-LDM** and provide the details in Algorithm 2 and Appendix B.5.

### 4.3 MULTI-STEP OPTIMIZATION

The current framework performs a one-step gradient descent on the clean data $x_{0|t}$ for the objective in equation 5. Nevertheless, for this objective, we can also employ more sophisticated optimization solvers such as nonlinear conjugate gradient method (Hager & Zhang, 2006), and provided the manifold remains preserved, execute multiple optimization iterations. This can potentially lead to improvements in both quality and speed.

Although motivated differently, "Time-Traveling" or "Repainting" (Lugmayr et al., 2022; Wang et al., 2022; Yu et al., 2023) is another technique that implicitly performs a multi-step optimization to minimize the guidance loss. Specifically, the process of adding noise after each gradient descent step can be interpreted as stochastic optimization via stochastic gradient Langevin dynamics (Welling & Teh, 2011). We show the results where we employ the both nonlinear conjugate gradient method and time-traveling for faceID guidance Stable Diffusion generation in Appendix E.5, Figure 18.

## 5 EXPERIMENTS

We empirically compare the performance of our proposed methods with baselines in three experimental settings. For the pixel domain diffusion, we test our methods with a simpler linear inverse problem and a more complex nonlinear problem. For the latent diffusion models, we evaluate our method with the two conditions at the same time, one included in the pre-trained model setting and the other provided by the loss function, to examine its ability to understand compositional conditions. We also provide further details of the experiments in Appendix D.

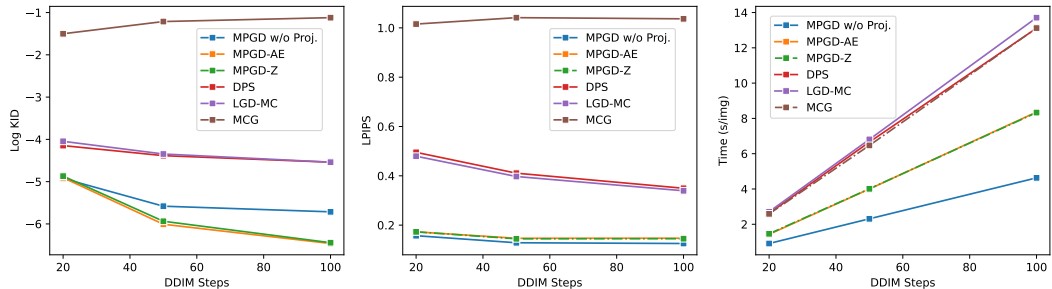

Figure 5: Quantitative results of FFHQ super-resolution experiment that compares fidelity (log KID), guidance quality (LPIPS) and inference time across different numbers of DDIM steps.

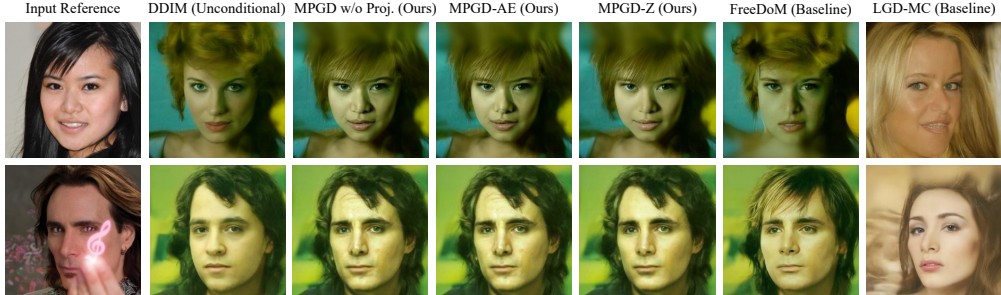

Figure 6: Examples of the FaceID guidance generation with pre-trained CelebA-HQ models.

## 5.1 PIXEL SPACE DIFFUSION MODELS

In this section, we evaluate the performance of our proposed pixel domain methods (i.e., **MPGD w/o Proj.**, **MPGD-AE**, and **MPGD-Z**) with two different sets of conditional image generation tasks: solving linear inverse problems and human face generation guided by face recognition loss, which we refer to as FaceID guidance generation. For MPGD-AE and MPGD-Z, we use the pre-trained VQGAN models provided by Rombach et al. (2021). To further demonstrate the applicability of our method, we add the results of CLIP-guided generation experiments in Appendix E.1.

### 5.1.1 NOISY LINEAR INVERSE PROBLEM

For linear tasks, we use noisy super-resolution and noisy Gaussian deblurring as the test bed. We choose DPS (Chung et al., 2023a), LGD-MC (Song et al., 2023b), and MCG (Chung et al., 2022) as the basleines. We test each method with two pre-trained diffusion models provided by Chung et al. (2023a): one trained on FFHQ dataset (Karras et al., 2019) and another on ImageNet (Deng et al., 2009), both with $256 \times 256$ resolution. Measurements in both tasks have a random noise with a variance of $\sigma^2 = 0.05^2$. We evaluate each task on a set of 1000 samples. We use the Kernel Inception distance (KID) (Bińkowski et al., 2018) to assess the fidelity, Learned Perceptual Image Patch Similarity (LPIPS) (Zhang et al., 2018) to evaluate the guidance quality, and the inference time to test the efficiency of the methods. All experiments are conducted on a single NVIDIA GeForce RTX 2080 Ti GPU. Figure 4 shows the generated examples for qualitative comparison, and Figure 5 presents the quantitative results for the super-resolution task on FFHQ. All three of our methods significantly outperform the baselines with all metrics tested across a variety of different numbers of DDIM steps, and we can observe manifold projection improves the sample fidelity by a large margin.

### 5.1.2 FACEID GUIDANCE

We also evaluate our proposed methods on the more challenging nonlinear task of FaceID guided human face image generation. The goal of this task is to generate facial images that resemble reference faces. We choose FreeDoM (Yu et al., 2023) and LGD-MC (Song et al., 2023b) as baseline methods. We test all methods with the pretrained diffusion model for the CelebA-HQ $256 \times 256$ dataset provided by Yu et al. (2023) and 50 DDIM steps. We generate 1000 facial images using the CelebA-HQ test set as reference images and evaluate the results using KID and FaceID Loss with a

Table 1: Quantitative results for CelebA-HQ 256×256 FaceID guidance experiment.

| Method | KID↓ | FaceID↓ | Time↓ |
|---|---|---|---|
| DDIM | 0.0442 | 1.3914 | 3.41s |
| FreeDoM | 0.0452 | 0.5690 | 10.65s |
| LGD-MC | 0.0448 | 0.6783 | 14.64s |
| MPGD | 0.0473 | **0.5163** | **5.82s** |
| MPGD-AE | 0.0467 | 0.5309 | 7.78s |
| MPGD-Z | **0.0445** | 0.5791 | 6.93s |

Table 2: Quantitative results for style guidance with Stable Diffusion experiment. Our method finds the sweet spot between following the prompt and following the style guidance.

| Method | Style↓ | CLIP↑ | Time↓ | VRAM↓ |
|---|---|---|---|---|
| DDIM | 761.0 | 31.61 | 13.89s | 10.80 GB |
| FreeDoM | 498.8 | 30.14 | 26.50s | 17.30 GB |
| LGD-MC | 404.0 | 21.16 | 37.43s | 31.65 GB |
| MPGD-LDM | 441.0 | 26.61 | **19.83s** | **15.53 GB** |

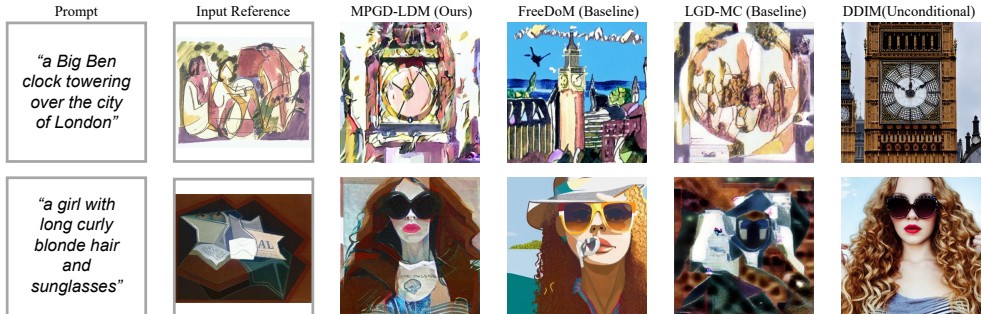

Figure 7: Qualitative results for text-to-image style guidance experiment with Stable Diffusion.

single NVIDIA GeForce RTX 3090 Ti GPU. Figure 6 shows the generated samples for qualitative comparison, and Table 1 presents the quantitative metrics. Our methods demonstrates comparable or superior sample quality with substantial speed-ups compared to the baselines. In addition, we also notice that our methods are able to maintain the overall geometry generated by DDIM and only make changes to the semantics that are relevant to the guidance. This observation suggests that our method is able to operate guidance in the tangent spaces of the DDIM samples.

## 5.2 LATENT DIFFUSION MODELS

To evaluate **MPGD-LDM**, we test our methods against the same baselines as the pixel-space FaceID experiments with text-to-image style guided generation task. The goal of this task is to generate images that fit both the text input prompts and the style of the reference images. We use Stable Diffusion (Rombach et al., 2021) as the pre-trained text-to-image model and deploy the guided sampling methods to incorporate a style loss, which is calculated by the Frobenius norm between the Gram matrices of the reference images and the generated images. For reference style images and text prompts, we randomly created 1000 conditioning pairs, using images from WikiArt (Saleh & Elgammal, 2015) and prompts from PartiPrompts (Yu et al., 2022) dataset. Figure 7 and Table 2 show qualitative and quantitative results for this experiment respectively. All the samples are generated on a single NVIDIA A100 GPU with 100 DDIM steps. Our method finds the sweet spot between following the text prompts, which usually instruct the generation to generate photo realistic images that do not suit the given style, and following the style guidance, which deviate from the prompts. Notably, because MPGD does not require propagation through the diffusion model, our method can provide significant speedup and can be fitted into a 16GB GPU while all the other methods cannot.

## 6 CONCLUSION

In this paper, we proposed Manifold Preserving Guided Diffusion (MPGD), a novel framework anchored in the manifold constraint within the diffusion generation process for conditional generation. By focusing on manifold preserving guidance, our approach promises high quality conditionally generated samples, while reducing the computational cost and memory, paving the way for more accessible and reliable guided generation processes. This approach leverages the pretrained autoencoders to ensure the manifold constraints, offering an efficient solution to the challenges in guided generation. Furthermore, our method incorporates the optimization strategies that enhance the effectiveness of the sampling process.

## 7 ETHICS STATEMENT

As a training-free guided generation method, our MPGD offers a way to approach low-resource control of the large scale pre-trained models. However, while MPGD facilitates low-cost human control over pre-trained models, our method is still subject to potential risks including biases, copyright issues and intentional malicious content generation that currently exist in large-scale pre-trained models. We acknowledge the importance of addressing these ethical considerations. Our commitment to ethical research practices extends to ensuring that our contributions do not exacerbate existing inequalities or perpetuate harmful biases. Upon the release of our code, we will implement safe guards to mitigate inappropriate content creation and we are dedicated to update our safe guarding system to keep up with the research regarding safer content generation in the future.

## 8 ACKNOWLEDGEMENT

This work is partially supported by ONRN000142312368. We would like to thank Zhengyang Geng, Ruitian Zhai, Martin Q. Ma, Jing Yu Koh and Ellie Haber for their helpful feedbacks.

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

## A  RELATED WORKS

**Methods that try to address the manifold-related issues**   Several papers (Chung et al., 2022; 2023b) have raised similar issues in the context of solving linear inverse problems using pre-trained diffusion models. In particular, Chung et al. (2023b) attempts to tackle the linear inverse problems by using the conjugate gradient method to maintain the samples on a linear data manifold. However, this solution defines the linear data manifold as Krylov subspace of the linear operator, which limits the applicability to linear inverse problems. Moreover, the Krylov subspace usually does not precisely align with the data manifold in many application scenarios and therefore their analysis does not generalize to many practical setting.

**Methods that require fine-tuning of pretrained models**   Prior to the introduction of the diffusion model, there were some methods that try to finetune Generative Adversarial Networks (GANs) for various tasks such as image-to-image translation (Isola et al., 2017; Zhu et al., 2017). Methods such as ControlNet (Zhang et al., 2023) and T2I-Adapter (Mou et al., 2023) are known for adding controllability of the model by finetuning pretrained diffusion models, for new conditional settings. Textual Inversion (Gal et al., 2022) DreamBooth (Ruiz et al., 2023) also requires finetuning with a small sef of images to customize (or personalize) generated images.

**Methods that don't require fine-tuning of pretrained models**   Using pre-trained model to address various tasks without additional training is an idea shared by many papers including Song et al. (2021b), SDEdit (Meng et al., 2022) and Repaint (Lugmayr et al., 2022). SNIPS (Kawar et al., 2021), DDRM (Kawar et al., 2022), DDNM (Wang et al., 2022), DPS (Chung et al., 2023a) and ΠGDM (Song et al., 2022) have expanded this framework to general linear inverse problems (specifically, DPS and ΠGDM also encompass nonlinear inverse problems) and have broadened their applicability. Additionally, methods such as PnP (Graikos et al., 2022) and RED-Diff (Mardani et al., 2023) achieve this objective by solving optimization problems that incorporate pre-trained diffusion models. Recently, UGD (Bansal et al., 2023), FreeDoM (Yu et al., 2023), and LGD (Song et al., 2023b) have been proposed to increase the range of tasks they can handle by making the design of the loss function more flexible. Attempts have also have been to apply latent diffusion models (LDM) within this framework, UGD and FreeDoM have enabled the use of LDM by incorporating the decoder into the loss function, leveraging the differentiability of the decoder. Moreover, papers (Song et al., 2023a; Rout et al., 2023; Fabian et al., 2023) focus on the utilization of latent diffusion models and address problems that arises in such situations.

## B  PROOFS AND THEORETICAL ANALYSIS

### B.1  PROOF OF PROPOSITION 1

**Proposition 1**   (Formal, Extended from Chung et al. (2022; 2023b)) *Define* $d(x, \nu, \mathcal{M}) := \inf_{x' \in \mathcal{M}} \|x - \nu x'\|_2$ *for* $\nu > 0$, *and* $B(\mathcal{M}; r) := \{x \in \mathbb{R}^d \mid d(x, 1, \mathcal{M}) < r\}$ *for* $r > 0$. *Consider the distribution of noisy data* $p_t(x_t) := \int p(x_t|x)p(x)dx$, *where* $p(x_t|x) := \mathcal{N}(\sqrt{\bar{\alpha}_t}x, (1 - \bar{\alpha}_t)I)$. *Then under Assumption 1.1,* $p_t(x_t)$ *is "probabilistically concentrated" on the* $(d - 1)$-*dimensional manifold* $\mathcal{M}_t$ *defined as*

$$\mathcal{M}_t := \{x \in \mathbb{R}^d \mid d(x, \sqrt{\bar{\alpha}_t}, \mathcal{M}) = \sqrt{(1 - \bar{\alpha}_t)(d - k)}\}.$$

*That is, for any* $0 < \delta \leq 1$, *there is an* $0 < \epsilon_{\delta, d-k} \leq 1$ *which is monotonically decreasing with respect to* $\delta$ *and* $(d - k)$ *such that*

$$\mathbb{P}(x_t \in B(\mathcal{M}_t; \epsilon_{\delta, d-k}\sqrt{(1 - \bar{\alpha}_t)(d - k)})) \geq 1 - \delta.$$

*Proof.* The proof follows Chung et al. (2022; 2023b) and here we provide an extended version. Without loss of generality, we define $\mathcal{M} = \{x \in \mathbb{R}^d | x_{k+1:d} = 0\}$ from the linear subspace manifold assumption. Let $X$ be a $\chi^2$ random variable with $l$ degrees of freedom. A concentration bound by Laurent & Massart (2000) implies that for all $\tau > 0$ we have

$$\mathbb{P}(X - l \geq 2\sqrt{l\tau} + 2\tau) \leq e^{-\tau}$$
$$\mathbb{P}(X - l \leq -2\sqrt{l\tau}) \leq e^{-\tau}. \tag{10}$$

Since $\sum_{i=k+1}^{d} x_{t,i}^2/(1 - \bar{\alpha}_t)$ is a $\chi^2$ random variable with $d - k$ degrees of freedom, by plugging into $\tau = (d - k)\epsilon'$ we have

$$\mathbb{P}\left(-2(d-k)\sqrt{\epsilon'} \leq \sum_{i=k+1}^{d} \frac{x_{t,i}^2}{1 - \bar{\alpha}_t} - (d-k) \leq 2(d-k)(\sqrt{\epsilon'} + \epsilon')\right)$$

$$= \mathbb{P}\left(\sqrt{\sum_{i=k+1}^{d} x_{t,i}^2} \in \left(r_t\sqrt{\max\{0, 1 - 2\sqrt{\epsilon'}\}}, r_t\sqrt{1 + 2\sqrt{\epsilon'} + 2\epsilon'}\right)\right)$$

$$\geq 1 - 2e^{-(d-k)\epsilon'}, \tag{11}$$

where $r_t := \sqrt{(1 - \bar{\alpha}_t)(d - k)}$. As a result, for any $0 < \delta \leq 1$, by setting

$$\epsilon'_{\delta,d-k} = -\frac{1}{d-k}\log\frac{\delta}{2}$$

$$\epsilon_{\delta,d-k} = \min\left\{1 - \sqrt{\max\{0, 1 - 2\sqrt{\epsilon'_{\delta,d-k}}\}}, \sqrt{1 + 2\sqrt{\epsilon'_{\delta,d-k}} + 2\epsilon'_{\delta,d-k}} - 1\right\}, \tag{12}$$

we have an $0 < \epsilon_{\delta,d-k} \leq 1$ such that

$$\mathbb{P}(x_t \in B(\mathcal{M}_t; \epsilon_{\delta,d-k}\sqrt{(1 - \bar{\alpha}_t)(d - k)})) \geq 1 - \delta.$$

$\epsilon_{\delta,d-k}$ is monotonically decreasing with respect to $\delta$ and $d-k$ since $\epsilon'_{\delta,d-k}$ is monotonically decreasing with respect to $\delta$ and $d - k$ and $\epsilon_{\delta,d-k}$ is monotonically increasing with respect to $\epsilon'_{\delta,d-k}$. $\square$

### B.2 PROOF OF THEOREM 1

First, we confirm the following lemmas.

**Lemma 1.** *(Total noise (Chung et al., 2023b)) Consider the optimality of the diffusion model, i.e., $\epsilon_\theta(\sqrt{\bar{\alpha}_t}x + \sqrt{1 - \bar{\alpha}_t}\epsilon_t, t) = \epsilon_t$ for $x \in \mathcal{M}$. For some $\tilde{\epsilon} \sim \mathcal{N}(0, I)$, the sum of noise components $\sqrt{1 - \bar{\alpha}_{t-1} - \sigma_t^2}\epsilon_\theta(x_t, t) + \sigma_t\epsilon_t$ in DDIM sampling (Equation 1) can expressed as*

$$\sqrt{1 - \bar{\alpha}_{t-1} - \sigma_t^2}\epsilon_\theta(x_t, t) + \sigma_t\epsilon_t = \sqrt{1 - \bar{\alpha}_{t-1}}\tilde{\epsilon}. \tag{13}$$

*Proof.* Since $\sqrt{1 - \bar{\alpha}_{t-1} - \sigma_t^2}\epsilon_\theta(x_t, t)$ and $\sigma_t\epsilon_t$ are independent, their sum is the sum of independent Gaussian random variables. Consequently, the resulting Gaussian distribution has a mean of 0 and a variance of $(1 - \bar{\alpha}_{t-1} - \sigma_t^2) + \sigma_t^2 = (1 - \bar{\alpha}_{t-1})$. $\square$

**Lemma 2.** *Let the data distribution $p(x)$ be a probability distribution with support on the linear manifold $\mathcal{M}$ that satisfies Assumption 1.1. For any $x \sim p(x)$, consider $x_{t-1} = \sqrt{\bar{\alpha}_{t-1}}x + \sqrt{1 - \bar{\alpha}_{t-1} - \sigma_t^2}\epsilon_\theta(x_t, t) + \sigma_t\epsilon_t$. Then its the marginal distribution $\hat{p}_{t-1}(x_{t-1})$, which is defined as*

$$\hat{p}_{t-1}(x_{t-1}) = \int \mathcal{N}(x_{t-1}; \sqrt{\bar{\alpha}_{t-1}}x + \sqrt{1 - \bar{\alpha}_{t-1} - \sigma_t^2}\epsilon_\theta(x_t, t), \sigma_t^2 I)p(x_t|x)p(x)dxdx_t \tag{14}$$

*is probabilistically concentrated on $\mathcal{M}_{t-1}$ for $\epsilon_t \sim \mathcal{N}(0, I)$, pre-trained optimal diffusion model noise estimator $\epsilon_\theta$, and its corresponding variance schedulers $\bar{\alpha}_t, \sigma_t$.*

*Proof.* By Lemma 1, the multivariate normal distribution in Equation 14 has a mean $\sqrt{\bar{\alpha}_{t-1}}x$ and a covariance matrix $(1 - \bar{\alpha}_{t-1})I$. Consequently, the marginal distribution of the target can be represented as

$$\hat{p}_{t-1}(x_{t-1}) = \int \mathcal{N}(x_{t-1}; \sqrt{\bar{\alpha}_{t-1}}x, (1 - \bar{\alpha}_{t-1})I)p(x)dx, \tag{15}$$

which is the same as the marginal distribution defined in Proposition 1. Therefore, in accordance with Proposition 1, the probability distribution $\hat{p}_{t-1}(x_{t-1})$ probabilistically concentrates on $\mathcal{M}_{t-1}$. $\square$

Finally, using both Lemma 1 and Lemma 2, we can prove Theorem 1. In the main text, we include the following informal statement.

**Theorem 1** *(Informal) Assume the gradient $\nabla_{x_{0|t}} L(x_{0|t}; y)$ lies on the tangent space $\mathcal{T}_{x_{0|t}} \mathcal{M}$, and the diffusion model $\epsilon_\theta(x_t, t)$ is optimal. Then with Assumption 1.1, scalar $c_t > 0$ and update rule*

$$x_{t-1} = \sqrt{\bar{\alpha}_{t-1}}(x_{0|t} - c_t \nabla_{x_{0|t}} L(x_{0|t}; y)) + \sqrt{1 - \bar{\alpha}_{t-1} - \sigma_t^2} \epsilon_\theta(x_t, t) + \sigma_t \epsilon_t, \tag{6}$$

*we can obtain an $x_{t-1}$ whose marginal distribution is probabilistically concentrated on $\mathcal{M}_{t-1}$.*

Here we also include and prove the formal statement below.

**Theorem 1** *(formal) Let the data distribution $p(x)$ be a probability distribution with support on the linear manifold $\mathcal{M}$ that satisfies Assumption 1.1 and $c_t > 0$ is a scalar function depending on $t$. Assume that the gradient $\nabla_{x_{0|t}} L(x_{0|t}; y)$ lies on the tangent space $\mathcal{T}_{x_{0|t}} \mathcal{M}$ for $x_{0|t} = \frac{1}{\sqrt{\bar{\alpha}_t}}(x_t - \sqrt{1 - \bar{\alpha}_t} \epsilon_\theta(x_t, t))$, and consider the diffusion model $\epsilon_\theta(x_t, t)$ is optimal. Let*

$$m_{t-1}(x_t) = \sqrt{\bar{\alpha}_{t-1}}(x_{0|t} - c_t \nabla_{x_{0|t}} L(x_{0|t}; y)) + \sqrt{1 - \bar{\alpha}_{t-1} - \sigma_t^2} \epsilon_\theta(x_t, t). \tag{16}$$

*Then for $x_{t-1} \sim \mathcal{N}(x_{t-1}; m_{t-1}(x_t), \sigma_t^2 I)$, that is,*

$$x_{t-1} = m_{t-1}(x_t) + \sigma_t \epsilon_t, \quad \epsilon_t \sim \mathcal{N}(\epsilon_t; 0, I) \tag{17}$$

*its marginal probability distribution*

$$\hat{p}_{m_{t-1}}(x_{t-1}) = \int \mathcal{N}(x_{t-1}; m_{t-1}(x_t), \sigma_t^2 I) p(x_t | x) p(x) dx dx_t. \tag{18}$$

*is probabilistically concentrated on $\mathcal{M}_{t-1}$.*

*Proof.* Firstly, we prove that for all $t$, there exists an $x \in \mathcal{M}$ such that the $x_t$ generated from Equation 17 can also be generated by the forward process of diffusion from $x$. In other words, $x_t = \sqrt{\bar{\alpha}_t} x + \sqrt{1 - \bar{\alpha}_t} \epsilon$ for $x \in \mathcal{M}, \epsilon \sim \mathcal{N}(0, I)$. We prove this by induction.

For the base case, let $t = T$. Since we use the same initial noisy sample $x_T$ from the Gaussian prior, trivially $x_T$ can also be expressed as $\sqrt{\bar{\alpha}_T} x + \sqrt{1 - \bar{\alpha}_T} \epsilon_T$ for some $x \sim p(x)$ by construction of the diffusion process. Since the support of $p(x)$ lies on $\mathcal{M}$, this $x$ is on the data manifold $\mathcal{M}$.

Now suppose for all $t \geq T_1$, there exists an $x \in \mathcal{M}$ such that $x_t = \sqrt{\bar{\alpha}_t} x + \sqrt{1 - \bar{\alpha}_t} \epsilon$ for $\epsilon \sim \mathcal{N}(0, I)$. Then since the diffusion model is optimal, $\epsilon_\theta(x_{T_1}, T_1) = \epsilon_\theta(\sqrt{\bar{\alpha}_{T_1}} x + \sqrt{1 - \bar{\alpha}_{T_1}} \epsilon, T_1) = \epsilon$. Therefore, $x_{0|T_1} = x \in \mathcal{M}$. Then, under the linear manifold hypothesis and considering the gradient $\nabla_{x_{0|T_1}} L(x_{0|T_1}; y)$ lies on the tangent space $\mathcal{T}_{x_{0|T_1}} \mathcal{M}$, for any $c_t > 0$, we can have $x' = (x_{0|T_1} - c_t \nabla_{x_{0|T_1}} L(x_{0|T_1}; y))$ is on $\mathcal{M}$, since the tangent space itself coincides with the manifold. By Lemma 1, we know that $\sqrt{1 - \bar{\alpha}_{T_1-1} - \sigma_{T_1}^2} \epsilon_\theta(x_{T_1}, T_1) + \sigma_{T_1} \epsilon_{T_1} = \sqrt{1 - \bar{\alpha}_{T_1-1}} \tilde{\epsilon}$ for some $\tilde{\epsilon} \sim \mathcal{N}(0, I)$. Therefore, $x_{T_1-1}$ can also be expressed as $\sqrt{\bar{\alpha}_{T_1-1}} x' + \sqrt{1 - \bar{\alpha}_{T_1-1}} \tilde{\epsilon}$ for $x' \in \mathcal{M}$. Hence complete the proof by induction.

Now that we have proved that for all $t$, there exists an $x \in \mathcal{M}$ such that the $x_t$ generated from Equation 17 can also be generated by the forward process of diffusion from $x$, we can directly apply Lemma 2, and hve the marginal distribution $\hat{p}_{m_{T-1}}(x_{T-1})$, as obtained by the update rule in Equation 17, is probabilistically concentrated on $\mathcal{M}_{T-1}$.

$\square$

Besides being on the manifold, the generated noisy samples should also reflect the guidance correctly. Here we theoretically verify the quality of the guidance by showing that samples obtained from our new update rule is in the vicinity of the samples obtained from the DPS update rule:

**Proposition 2.** *With the same assumptions and notations as Theorem 1, for certain given $x_t, \epsilon_t$, denote*

$$x_{t-1}^{(MPGD)} = \sqrt{\bar{\alpha}_{t-1}}(x_{0|t} - c_t \nabla_{x_{0|t}} L(x_{0|t}; y)) + \sqrt{1 - \bar{\alpha}_{t-1} - \sigma_t^2} \epsilon_\theta(x_t, t) + \sigma_t \epsilon_t$$

*to be the updated sample obtained from Equation 6 and*

$$x_{t-1}^{(DPS)} = \sqrt{\bar{\alpha}_{t-1}} x_{0|t} + \sqrt{1 - \bar{\alpha}_{t-1} - \sigma_t^2} \epsilon_\theta(x_t, t) + \sigma_t \epsilon_t - \rho_t \nabla_{x_t} L(x_{0|t}, y)$$

to be the updated sample obtained from Equation 4. If $\|\frac{\partial L}{\partial x_{0|t}}\frac{\partial \epsilon_\theta(x_t,t)}{\partial x_t}\|$ is upper bounded by small positive constant $\kappa$, then with some $c_t > 0$, the distance between $x_{t-1}^{(MPGD)}$ and $x_{t-1}^{(DPS)}$ is upper bounded by constant $\kappa\rho_t\frac{\sqrt{1-\bar{\alpha}_t}}{\sqrt{\bar{\alpha}_t}}$. In other words,

$$\|x_{t-1}^{(DPS)} - x_{t-1}^{(MPGD)}\| \leq \kappa\rho_t\frac{\sqrt{1-\bar{\alpha}_t}}{\sqrt{\bar{\alpha}_t}}$$

*Proof.* By chain rule, we know that

$$\nabla_{x_t}L(x_{0|t},y) = \frac{\partial L}{\partial x_{0|t}}\frac{1}{\sqrt{\bar{\alpha}_t}}(I - \sqrt{1-\bar{\alpha}_t}\frac{\partial \epsilon_\theta(x_t,t)}{\partial x_t})$$
$$= \frac{1}{\sqrt{\bar{\alpha}_t}}\frac{\partial L}{\partial x_{0|t}} - \frac{1}{\sqrt{\bar{\alpha}_t}}\sqrt{1-\bar{\alpha}_t}\frac{\partial L}{\partial x_{0|t}}\frac{\partial \epsilon_\theta(x_t,t)}{\partial x_t}$$
$$= \frac{1}{\sqrt{\bar{\alpha}_t}}\nabla_{x_{0|t}}L(x_{0|t},y) - \frac{\sqrt{1-\bar{\alpha}_t}}{\sqrt{\bar{\alpha}_t}}\frac{\partial L}{\partial x_{0|t}}\frac{\partial \epsilon_\theta(x_t,t)}{\partial x_t}$$

Since $\|\frac{\partial L}{\partial x_{0|t}}\frac{\partial \epsilon_\theta(x_t,t)}{\partial x_t}\|$ is upper bounded by some constant $\kappa$, we can have

$$\|\frac{1}{\sqrt{\bar{\alpha}_t}}\nabla_{x_{0|t}}L(x_{0|t},y) - \nabla_{x_t}L(x_{0|t},y)\| \leq \frac{\sqrt{1-\bar{\alpha}_t}}{\sqrt{\bar{\alpha}_t}}\kappa \qquad (19)$$

As a result, for $c_t = \frac{\rho_t}{\sqrt{\bar{\alpha}_{t-1}\bar{\alpha}_t}} > 0$

$$\|x_{t-1}^{(DPS)} - x_{t-1}^{(MPGD)}\|$$
$$= \|\sqrt{\bar{\alpha}_{t-1}}x_{0|t} + \sqrt{1-\bar{\alpha}_{t-1}-\sigma_t^2}\epsilon_\theta(x_t,t) + \sigma_t\epsilon_t - \rho_t\nabla_{x_t}L(x_{0|t};y)$$
$$- \left(\sqrt{\bar{\alpha}_{t-1}}(x_{0|t} - c_t\nabla_{x_{0|t}}L(x_{0|t};y)) + \sqrt{1-\bar{\alpha}_{t-1}-\sigma_t^2}\epsilon_\theta(x_t,t) + \sigma_t\epsilon_t\right)\|$$
$$= \|\sqrt{\bar{\alpha}_{t-1}}x_{0|t} - \rho_t\nabla_{x_t}L(x_{0|t};y) - \sqrt{\bar{\alpha}_{t-1}}(x_{0|t} - c_t\nabla_{x_{0|t}}L(x_{0|t};y))\|$$
$$= \|c_t\sqrt{\bar{\alpha}_{t-1}}\nabla_{x_{0|t}}L(x_{0|t};y) - \rho_t\nabla_{x_t}L(x_{0|t};y)\|$$
$$= \|\frac{\rho_t}{\sqrt{\bar{\alpha}_{t-1}\bar{\alpha}_t}}\sqrt{\bar{\alpha}_{t-1}}\nabla_{x_{0|t}}L(x_{0|t};y) - \rho_t\nabla_{x_t}L(x_{0|t};y)\|$$
$$= \rho_t\|\frac{1}{\sqrt{\bar{\alpha}_t}}\nabla_{x_{0|t}}L(x_{0|t};y) - \nabla_{x_t}L(x_{0|t};y)\|$$
$$\leq \kappa\rho_t\frac{\sqrt{1-\bar{\alpha}_t}}{\sqrt{\bar{\alpha}_t}}$$

$\square$

Batzolis et al. (2022) shows that as $t$ decreases, the score, i.e., $\epsilon_\theta(x_t,t)$, becomes perpendicular to the clean data manifold in practice. As a result, if $\frac{\partial L}{\partial x_{0|t}}$ is on the tangent space $\mathcal{T}_{x_{0|t}}\mathcal{M}$, $\|\frac{\partial L}{\partial x_{0|t}}\frac{\partial \epsilon_\theta(x_t,t)}{\partial x_t}\| \to 0$ as $t$ decreases. And therefore, empirically the upper bound constant $\kappa$ is very close to 0 when $t$ is small. Hence, in practice, our method can provide updated samples that reflect similar guidance to the ones from DPS while having marginal distributions that are probabilistically concentrated on the correct manifolds.

### B.3 PROOF OF THEOREM 2

To prove Theorem 2, we examine the following Lemmas 3 and 4.

**Lemma 3.** *Let $E$, $D$ be the encoder and decoder, respectively, of a perfect autoencoder for a data support $\mathcal{X} \subset \mathcal{M}$. For any $x_0 \in \mathcal{X}$, it holds that $x_0 = D(z_0)$, where $z_0 = E(x_0)$. Then, the Jacobian $\frac{\partial E}{\partial x_0}$ of the encoder evaluated at $x_0$ and the Jacobian $\frac{\partial D}{\partial z_0}$ of the decoder evaluated at $z_0$ satisfy $\frac{\partial E}{\partial x_0} \frac{\partial D}{\partial z_0} = I$, where $I$ is the identity matrix.*

*Proof.* Given an encoder $E$ and a decoder $D$, for a certain $z_0$ in $\mathcal{Z}$, it holds that $z_0 = E(D(z_0))$. For convenience, we denote $x_0 = D(z_0)$. Differentiating both sides of $z_0 = E(D(z_0))$ with respect to $z_0$, we obtain:

$$I = \frac{\partial E}{\partial x_0} \frac{\partial D}{\partial z_0}. \tag{20}$$

$\square$

**Lemma 4.** *With perfect autoencoder and Lemma 3, $\frac{\partial E}{\partial x_0}^\top$ and $\frac{\partial D}{\partial z_0}$ share the same range. In other words, the subspaces spanned by the column vectors of both matrices are identical.*

*Proof.* We aim to show that the image spaces of $\frac{\partial E}{\partial x_0}^\top$ and $\frac{\partial D}{\partial z_0}$ are identical. By Lemma 3, $\frac{\partial E}{\partial x_0} \frac{\partial D}{\partial z_0} = I$. Let the row vectors of $\frac{\partial E}{\partial x_0}$ be denoted as $v_{E,1}^t, \ldots, v_{E,k}^t$ and the column vectors of $\frac{\partial D}{\partial z_0}$ as $v_{D,1}, \ldots, v_{D,k}$. It holds that $v_{E,i}^t v_{D,i} = 1$ and for $i \neq j$, $v_{E,i}^t v_{D,j} = 0$.

Now, considering any column vector $v_{D,i}$ of $\frac{\partial D}{\partial z_0}$, it can be expressed in terms of the row vectors of $\frac{\partial E}{\partial x_0}$ as $v_{D,i} = v_{E,i} \times \frac{\|v_{D,i}\|_2}{\|v_{E,i}\|_2}$. This implies that any element of the subspace spanned by the column vectors of $\frac{\partial D}{\partial z_0}$ can be expressed as a linear combination of the row vectors of $\frac{\partial E}{\partial x_0}$. Conversely, the same holds true. Therefore, the subspace spanned by the column vectors of $\frac{\partial D}{\partial z_0}$ coincides with the subspace spanned by the row vectors of $\frac{\partial E}{\partial x_0}$. In conclusion, the image spaces of $\frac{\partial E}{\partial x_0}^\top$ and $\frac{\partial D}{\partial z_0}$ are identical. $\square$

**Theorem 2** *If an autoencoder with encoder $E$ and decoder $D$ is a perfect autoencoder for the support $\mathcal{X} \subset \mathcal{M}$ of the data distribution, then $\nabla_{x_0} L(D(E(x_0)); y) = \frac{\partial L}{\partial D} \frac{\partial D}{\partial E} \frac{\partial E}{\partial x_{0|t}} \in \mathcal{T}_{x_0}\mathcal{M}$.*

*Proof.* As stated in Shao et al. (2018), given the assumption of a perfect autoencoder, for any $z_0 \in \mathbb{R}^k$, the Jacobian $\frac{\partial D}{\partial z_0}$ maps the tangent space $\mathcal{T}_{z_0}\mathcal{Z}$ at $z_0$ to the tangent space $T_{x_0}\mathcal{M}$ of the data manifold at $x_0 = D(z_0)$. In other words, the range of the Jacobian $\frac{\partial D}{\partial z_0}$ lies within the tangent space at $x_0$. Since $\mathcal{Z} = \mathbb{R}^k$, its tangent spaces are isomorphic to $\mathbb{R}^k$. Therefore, for any vector $v_z \in \mathbb{R}^k$, the vector $\frac{\partial D}{\partial z_0} v_z$ lies in the tangent space at $x_0$. This means that when taking the gradient of the loss function with respect to $z_0$ and applying the Jacobian $\frac{\partial D}{\partial z_0}$ to the gradient, the resulting vector $\frac{\partial D}{\partial z_0} \frac{\partial D}{\partial z_0}^\top \frac{\partial L}{\partial x_0}^\top$ also lies in the tangent space at $x_0$. Finally, by Lemma 4, since $\frac{\partial E}{\partial x_0}^\top$ and $\frac{\partial D}{\partial z_0}$ share the same range, $\nabla_{x_0} L(D(E(x_0)), y) = \frac{\partial L}{\partial D} \frac{\partial D}{\partial E} \frac{\partial E}{\partial x_{0|t}} = (\frac{\partial E}{\partial x_0}^\top \frac{\partial D}{\partial z_0}^\top \frac{\partial L}{\partial x_0}^\top)^\top$ lies in the tangent space at $x_0$. $\square$

### B.4 THEORETICAL ANALYSIS ON MPGD-Z

In this section, we provide the theoretical analysis and proof for algorithm MPGD-Z as a proposition to Theorem 2.

**Proposition 3.** *If an autoencoder with encoder $E$ and decoder $D$ is a perfect autoencoder for $\mathcal{X} \subset \mathcal{M}$, then $D(\nabla_{z_0} L(D(z_0)); y) = D(\frac{\partial L}{\partial D} \frac{\partial D}{\partial z_0}) \in \mathcal{T}_{x_0}\mathcal{M}$.*

*Proof.* Similar to Theorem 2, by Lemma 4 $\frac{\partial D}{\partial z_0} = (\frac{\partial E}{\partial x_0})^\top$. Therefore, $\nabla_{z_0} L(D(z_0)); y) = \frac{\partial L}{\partial D} \frac{\partial D}{\partial z_0} = \frac{\partial L}{\partial D}(\frac{\partial E}{\partial x_0})^\top \in \mathcal{T}_{z_0}\mathcal{Z} \subset \mathcal{Z}$. Because we have linear manifold assumption, the updated $z_{0|t} = z_{0|t} + \nabla_{z_0} L(D(z_0)); y)$ is also on $\mathcal{Z}$. Since $D$ is surjective to $\mathcal{M}$, $D(z_0 + \nabla_{z_0} L(D(z_0)); y)$ is on the data manifold. $\square$

Hence the update rules for MPGD-Z is on-manifold.

Notice that in practice the autoencoder can exhibit reconstruction error. To mitigate this problem, we add the inference time reconstruction error $x_{0|t} - D(E(x_{0|t}))$ back to the guided clean data estimation after the update. In other words, we use the empirical update rule

$$z_{0|t} = E(x_{0|t})$$
$$\Delta x_{0|t} = x_{0|t} - D(z_{0|t})$$
$$z_{0|t} = z_{0|t} - c_t \nabla_{z_{0|t}} L(D(z_{0|t}); y)$$
$$x_{0|t} = D(z_{0|t}) + \Delta x_{0|t}$$

This empirical update rule can be viewed as adding a weighted regularization term $\lambda_{\text{rec}} \| x_{0|t} - \text{SG}(D(E(x_{0|t}))) \|^2$ to the guidance loss where $\lambda_{\text{rec}}$ is a scalar weight and $\text{SG}(D(E(x_{0|t})))$ is the reconstructed clean data estimation that is fixed before any guidance update (SG here denotes "stop gradient").

### B.5 THEORETICAL ANALYSIS ON MPGD-LDM

In this section, we provide the theoretical analysis for algorithm MPGD-LDM with perfect autoencoder assumption.

**Proposition 4.** *If an autoencoder with encoder $E$ and decoder $D$ is a perfect autoencoder for $\mathcal{X} \subset \mathcal{M}$, then a guided latent diffusion sampling described in Algorithm 2 can generate a sample $x_0 \in \mathcal{M}$.*

*Proof.* Since we have a perfect autoencoder, the latent space is exactly $\mathbb{R}^k$. As a result, the latent diffusion process will not move the latent sample out of the latent space. Because $D$ is surjective to the manifold, $D(z_0)$ is on the data manifold. □

## C EMPIRICAL VERIFICATION OF THE MANIFOLD PRESERVING ABILITIES

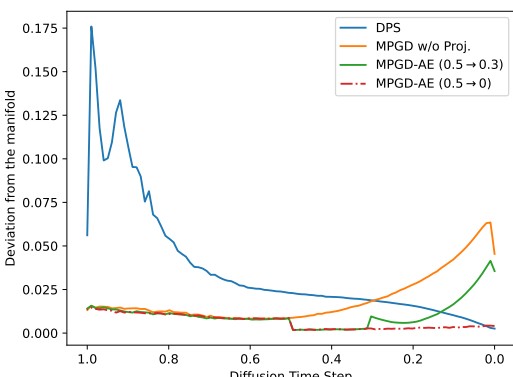

Figure 8: We analyze the deviation from the manifold throughout the diffusion process for different methods using the inner products between normalized score and the Jacobians from the guidance loss function.

In Figure 3, which we also include in this section as Figure 8, we empirically verifies VQGAN's manifold preserving ability by using it as the manifold projection function of MPGD-AE.

We use the diffusion model predicted score as a first-order Taylor series approximation of the log likelihood and calculate the inner product between the normalized score and the normalized Jacobian of the guidance loss as an indicator of how much the guidance deviate the intermediate samples from the original distribution, i.e., off the manifolds.

As a comparison, we also show the inner product curve for the baseline DPS and MPGD without manifold projection. We witness significant deviations in DPS at the beginning of the sampling

process and moderate ones in MPGD at the end. When applying VQGAN in diffusion time steps $t = 0.5$ to $t = 0$ (denoted as "MPGD-AE $(0.5 \rightarrow 0)$" in the plot), we can observe that the manifold projection effectively eliminates the deviation as the inner products become close to $0$.

Empirically, we only apply autoencoder projection for $t = 0.5$ to $t = 0.3$ (denoted as "MPGD-AE $(0.5 \rightarrow 0.3)$") for efficiency purpose, but our method is still able to produce high quality samples that follow the guidance.

## D  DETAILS ON EXPERIMENTS

### D.1  LINEAR CASE

**Baselines**  We employ DPS (Chung et al., 2023a), LGD-MC (Song et al., 2023b), and MCG (Chung et al., 2022) as baseline methods. Both DPS and LGD-MC approximate the log likelihood for noisy data to that of clean data, which is similar to our approach. MCG introduces a technique to correct intermediate samples from the generative process that deviate from the manifold in linear inverse problem settings.

**Experiment Setting and Datasets**  We evaluate our approach to the super-resolution task and the Gaussian deblurring task. In both experiments, the measurement process is given by $y = \mathcal{A}x + z$, where $\mathcal{A}$ is a known linear operator and $z$ is the measurement noise. The objective is to estimate the original data $x$ from the measurement $y$. We assume that the measurement noise is Gaussian in both cases. The log-likelihood for the clean data can be represented as $L(x; y) = \gamma \|y - \mathcal{A}x\|_2^2$ where $\gamma$ is a constant value, which we use as the loss function. More specifically, for the super-resolution task, the linear operator consists of a bicubic downsampling operator, which downsamples $256 \times 256$ images to $64 \times 64$. The variance of measurement noise is $0.05^2$. For Gaussian deblurring, the linear operator is a convolution operator with a $61 \times 61$ Gaussian blur kernel with an intensity value of 3.0. The variance of the measurement noise is also set to $0.05^2$.

We evaluate our approach using the FFHQ $256 \times 256$ (Karras et al., 2019) and ImageNet $256 \times 256$ (Deng et al., 2009) datasets. For FFHQ, we utilize a pretrained model from Choi et al. (2021) [1], and for ImageNet, we employ a pretrained model from Dhariwal & Nichol (2021) [2]. For all methods, including the proposed method, the same pre-trained models are used for each dataset.

For all methods, the number of DDIM steps is tested in three cases: $[20, 50, 100]$. The parameter $\eta$ is set to 0.5. The weight parameter scheduling is based on the implementation of DPS. The guidance weight hyperparameters for all of MPGD w/o proj., MPGD-AE, and MPGD-Z are 20,10,5 for DDIM steps 20, 50, 100 respectively. The weights for DPS is 0.3 as their default, for MCG is 100.0 and for LGD is 0.05 for the best empirical results we obtain. We follow the super-resolution setting in the LGD paper for its additional weight scheduling. The number of Monte Carlo samples for LGD is set to 10.

**Evaluation Metrics**  For each dataset, we perform inference using 1000 images from the test set. As evaluation metrics, we use the Kernel Inception distance (KID) (Bińkowski et al., 2018) to asses the fidelity, Learned Perceptual Image Patch distance (LPIPS) (Zhang et al., 2018) to evaluate guidance quality, and the inference time to test efficiency of the method. For linear cases, all the experiments are conducted on a single NVIDIA GeForce RTX 2080 Ti GPU. The inference time is measured by averaging the time taken to generate 20 images. During this evaluation, the batch size is set to 1.

### D.2  NONLINEAR CASE: DATA DOMAIN DIFFUSION MODELS

**Baselines**  As baselines that can solve general tasks using pretrained models, FreeDoM and LGD-MC are compared. FreeDoM and LGD have similar ideas to DPS, using a loss for clean data to approximate the log-likelihood for noisy data.

---

[1] https://github.com/jychoi118/ilvr_adm
[2] https://github.com/openai/guided-diffusion

**Experiment setting and Datasets**   The objective of FaceID-guided face image generation is to generate facial images that resemble reference faces. As with FreeDoM, we use a pretrained human face recognition network (Deng et al., 2019) to extract facial features. Specifically, we calculate the $\ell_2$ distance between the facial features extracted from the $x_{0|t}$ and those from the reference face image.

We test all methods with the pretrained diffusion model for the CelebA-HQ $256 \times 256$ dataset provided by Yu et al. (2023) [3] and 50 DDIM steps. All the samples are generated on a single NVIDIA RTX3090 GPU. $\eta$ is set to $0.5$. The weight parameter scheduling is based on the implementation of FreeDoM. We set the guidane weights to the value of 0.015, 0.015, 0.015, 100, and 50, for MPGD w/o proj., MPGD-AE, MPGD-Z, FreeDoM, and LGD, respectively. The number of Monte Carlo for LGD is set to 3, and the Monte Carlo parameter $r_t$ is set to $0.1\sqrt{1 - \bar{\alpha}_t}$.

**Evaluation Metrics**   We generate 1000 facial images using the CelebA-HQ test set as reference images and evaluate the results using KID and FaceID Loss. The inference time is measured by averaging the time taken to generate 10 images. During the inference, the batch size is set to 1.

### D.3   NONLINEAR CASE: LATENT DIFFUSION MODELS

**Baselines**   As baselines, we compare the proposed method with FreeDoM and LGD, similar to the case of data domain diffusion models.

**Experiment setting and Datasets**   We have the evaluation on the text-to-image style guided generation task, where the goal is to generate images that fit both the text input prompts and the style of the reference images. As the pretrained diffusion model, we use the `stable-Diffusion-v-1-4` checkpoint Rombach et al. (2021) [4]. The loss function involves calculating the Gram matrices (Johnson et al., 2016) of the intermediate layers of the CLIP image encoder for both the generated images and the reference style images, then using their Frobenius norms as the objective. More specifically, for a reference style image $x_{\text{ref}}$ and a decoded image $D(z_{0|t})$ from the estimated clean latent variable $z_{0|t}$, we compute the Gram matrices $G(x_{\text{ref}})_j$ and $G(D(z_{0|t}))_j$ corresponding to the features of the $j$-th layer of the image encoder. The loss function is then calculated as follows:

$$L(z_{0|t}; x_{\text{ref}}) = \|G(x_{\text{ref}})_j - G(D(z_{0|t}))_j\|_F^2, \tag{21}$$

where $\| \cdot \|_F^2$ denotes the Frobenius norm of a matrix. We adopt the third layer's features, consistent with the configuration used in FreeDoM. All the samples are generated on a single NVIDIA A100 GPU with 100 DDIM steps. $\eta$ is set to $1.0$. The weight parameter scheduling is based on the implementation of FreeDoM. We set the parameter $\rho$ to the values of 17.5, 0.2, and 15.0 for MPGD-LDM, FreeDoM, and LGD, respectively. Additionally, we configure the classifier-free guidance scale parameter to the value of 7.5, 5.0, and 5.0 for MPGD-LDM, FreeDoM, and LGD, respectively.

**Evaluation Metrics**   We use Style Score and CLIP score for evaluation. For reference style images and text prompts, we randomly created 1000 conditioning pairs, using images from WikiArt Saleh & Elgammal (2015) [5] and prompts from PartiPrompts Yu et al. (2022) dataset. The inference time is measured by averaging the time taken to generate 5 images. During the inference, the batch size is set to 1.

## E   ADDITIONAL RESULTS

### E.1   CLIP GUIDED GENERATION WITH PIXEL-SPACE CELEBA-HQ MODEL

To further demonstrate the applicability of our method, we conduct another experiment where we use a pre-trained CLIP model and text prompt to guide the generation of human faces using the pixel-space CelebA-HQ model, which is the same model used in the FaceID guided generation experiment. We use the $\ell_2$ Euclidean distance between the provided text prompts and the images

---

[3] `https://github.com/vvictoryuki/FreeDoM`
[4] `https://huggingface.co/CompVis/stable-diffusion-v-1-4-original`
[5] `https://www.wikiart.org/`

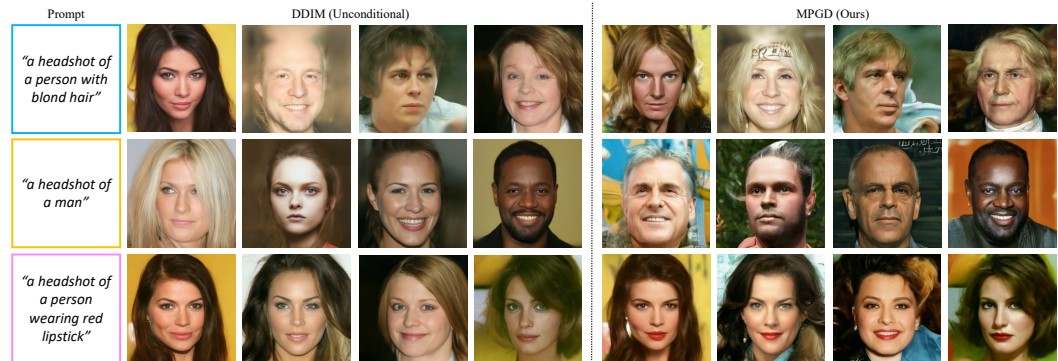

Figure 9: CLIP guided generation with Pixel-space CelebA-HQ Model

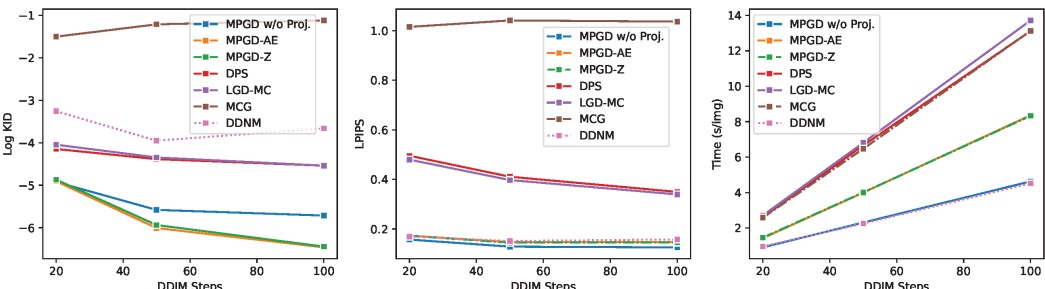

Figure 10: Quantitative comparison between our methods, DDNM and other baselines.

as the guidance loss. We also sample all images with 50 DDIM steps with $\eta = 1.0$. Images with prompt "a headshot of a person with blond hair" and "a headshot of a man" are generated with MPGD-Z, and images with prompt "a headshot of a person wearing red lipstick" is generated with MPGD-AE. For other hyper-parameters such as $\rho$ and time traveling steps, different prompts require different choices, which we have detailed discussions in later sections. In general, we find $\rho \in [1, 3]$ and less than 10 steps of time traveling for only a subset of diffusion step (similar to FreeDoM) to work well.

In Figure 9, we exhibit examples of samples guided by the text prompt, compared with unconditional DDIM samples generated from identical random seeds. Our method is able to create images that follow the text description provided while maintaining high fidelity.

## E.2 COMPARISON WITH DDNM

In this section, we compare our method with one of the state-of-the-art diffusion based inverse problem solver, Denoising Diffusion Null-space Model, DDNM (Wang et al., 2022) on the super-resolution task on the FFHQ dataset. Before we dive into the discussion about the experiment, we would like to emphasize that DDNM is designed for only solving linear inverse problems, and it requires direct access to the operation matrix, its pseudo-inverse/SVD and the noise scale for the measurement, which are not parts of the assumptions that we have in our problem setting. As a result, DDNM is not applicable to the general setting of paper. Nevertheless, we also think it is valuable to better position our paper in the literature of linear inverse problem solving, and therefore we conducted the experiments described below.

Table 3: PSNR comparison between DDNM and MPGD-Z.

| Method | PSNR ↑ | | |
| --- | --- | --- | --- |
| | DDIM Step = 20 | DDIM Step = 50 | DDIM Step = 100 |
| DDNM | 27.53 | 29.38 | 29.47 |
| MPGD-Z | 25.40 | 25.40 | 24.97 |

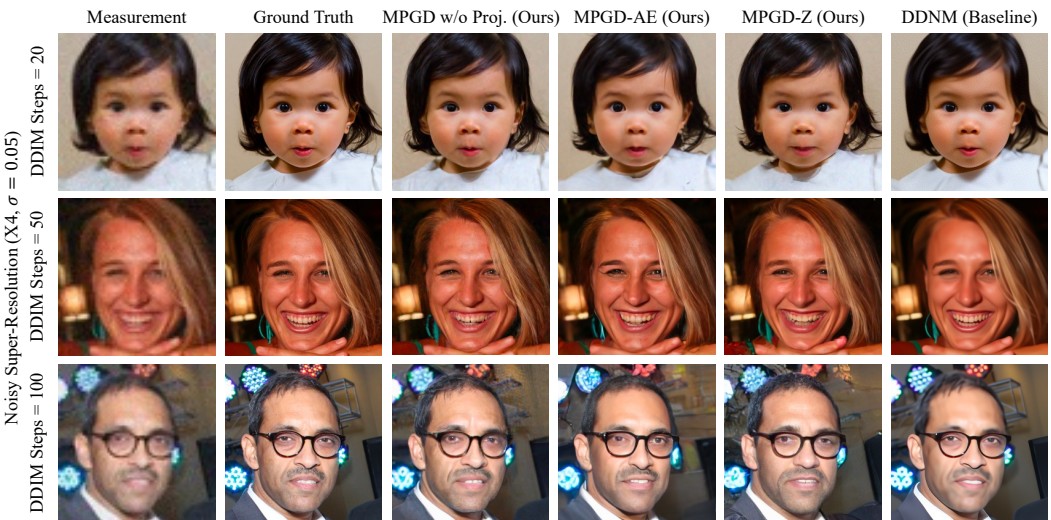

Figure 11: Qualitative comparison between our methods, DDNM and other baselines.

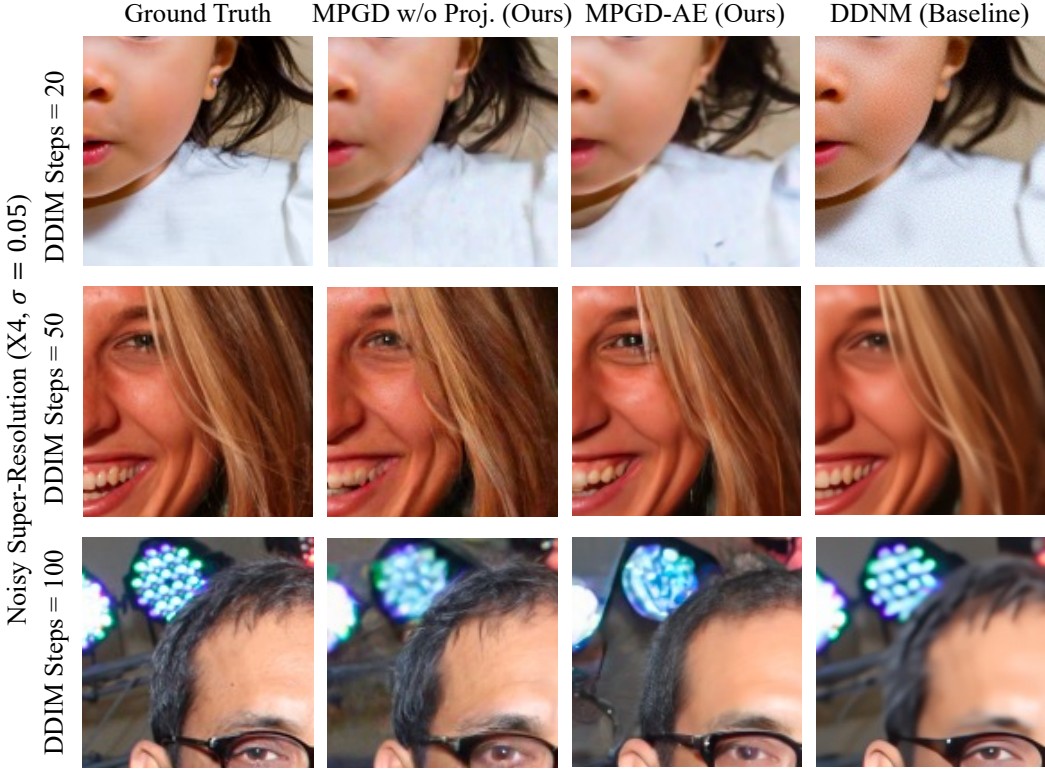

Figure 12: Detailed enlargement of the generated images produced by our methods in comparison with the ones produced by DDNM.

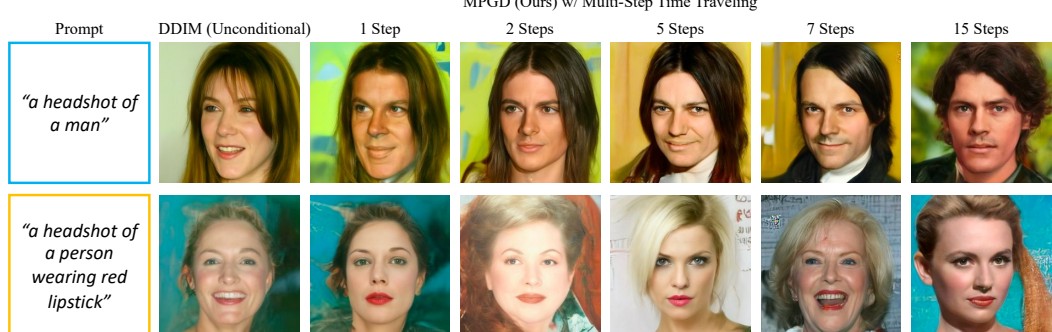

Figure 13: Qualitative showcase of the effect of different number of time traveling steps used in CLIP-guidance pixel-space CelebA-HQ model generation.

To make a fair comparison, we use the simplified version of DDNM with no time traveling and the same unconditional diffusion model pre-trained by the authors of DPS, which is a smaller model compared to the one DDNM used in their paper. Due to code availability, we use average pooling as the interpolation method, which is different from our original experiment setting.

Figures 11, 12, and 10 illustrate the qualitative results, detailed enlargements, and quantitative outcomes, respectively. As we observe in the figures, the simplified version of DDNM achieves an equally fast sampling speed compared to our method and obtains similar guidance quality. However, the images generated from DDNM exhibit various artifacts, such as high-frequency circular patterns and overly smooth generation. These artifacts prevent DDNM from maintaining high fidelity while our method can generate more realistic details.

Despite these findings, it is worth noting that the images generated from DDNM tend to maintain better shapes, whereas our method hallucinates small details more than DDNM. Consequently, regarding Peak Signal-to-noise ratio (PSNR) values, DDNM significantly outperforms our approach (Table 3). We also acknowledge that in the original paper of DDNM, in order to solve the noisy inverse problems, the authors suggest using multi-step time traveling to improve the performance, which we did not deploy in order to make a fair comparison in terms of run time. Therefore, DDNM still has certain advantages over our method in terms of solving specific inverse problems. We believe that integrating the consistency constraint from DDNM with our approach could potentially strengthen both methods, presenting a promising direction for future research.

### E.3 IMPACT OF THE NUMBER OF OPTIMIZATION STEPS

We explore the impact of varying optimization steps on generated samples. To illustrate, we first consider the time-traveling algorithm as performing more Langevin steps in one step, thereby increasing the likelihood of reaching an optimal solution in terms of loss, especially when these optima are significantly from the starting point.

For instance, in the task of sampling "a headshot of a man" using CLIP guidance and the pixel space CelebA-HQ model, if our initial unconditional DDIM sample produces a headshot of a woman, implementing a multi-step optimization process proves to be more advantageous. This is because the samples aligning more closely with the prompt are likely to be farther from the original unconditional sample. Conversely, for tasks that require only minor modifications, such as generating "a headshot of a person wearing red lipstick," achieving high quality is possible with fewer optimization steps. The generated images with various number of steps are provided in Figure 13. Overall, our results suggest that the more challenging the task (i.e., the greater the deviation required to reach the desired result in expectation), the more beneficial multi-step optimization becomes.

That being said, we do observe in practice that a large number of steps does not always benefit the generation. For example, we can start to observe unnatural artifacts appearing in the background of the image when sampling with 7 and 15 steps in the "red lipstick" experiment. In fact, we hypothesize that with a step size that is not infinitesimally small, asymptotically infinite-step optimization may lead to significant deviation from the data distribution.

Additionally, we explore the use of various optimization algorithms, such as nonlinear conjugate gradient, and the application of multi-step optimization to select subsets of steps in line with the FreeDoM framework. We also observe that step sizes need to be adjusted according to the number of steps used. The asymptotic behavior of the multi-step optimization and selecting appropriate hyper-parameters for these variations are promising areas for future research.

### E.4 INFLUENCE OF THE CLASSIFIER-FREE GUIDANCE SCALE IN STYLE GUIDANCE GENERATION EXPERIMENT

Table 4: Influence of the classifier-free guidance (CFG) scale on the style score and the CLIP score in MPGD-LDM style guided generation experiment.

| CFG Scale | Style ($\downarrow$) | CLIP ($\uparrow$) |
|:---:|:---:|:---:|
| 2.5 | 493.5 | 26.98 |
| 5.0 | 459.8 | **27.08** |
| 7.5 | **441.0** | 26.61 |

We expand our analysis to include a quantitative comparison of style-guided Stable Diffusion generation with various classifier-free guidance (CFG) scales. The parameter $\rho$ is set to $17.5$, while the CFG scale is selected from the set $[2.5, 5.0, 7.5]$. Table 4 shows the influence of the CFG scale on the style score and the CLIP score.

It's worth noting that the CFG scale has a positive impact on the style score, and strong CFG scales appear to help decrease the loss function. However, although in vanilla text-to-image generation tasks a larger CFG scale tends to lead to a higher CLIP score (Nichol et al., 2022), since the distribution we aim to sample from is also guided by another external loss function, we do not observe the same trend in our setting. In practice, we would suggest our users to adjust this hyperparameter to suit their preference of tradeoff between the style guidance and the text prompt condition.

### E.5 ADDITIONAL QUALITATIVE RESULTS IN THE EXPERIMENTS

We provide diverse additional qualitative results in Figure 14,15,16,17,18,19,21.

### E.6 USER STUDY

Table 5: User study results on style guidance Stable Diffusion generation task. "Style", "Text" and "Overall" represent style consistency, text prompt consistency and overal user preference. "(W/L/D)" represents the win/lose/draw ratios.

| Method | Style (W/L/D) | Text (W/L/D) | Overall (W/L/D) |
|:---|:---:|:---:|:---:|
| MPGD-LDM v.s. FreeDoM | 47%/45%/8% | 32%/66%/2% | 49%/45%/6% |
| MPGD-LDM v.s. LGD-MC | 27%/64%/9% | 69%/29%/2% | 53%/43%/4% |

We conduct a user study for the style-guided Stable Diffusion generation task on Amazon Mechanical Turk to compare our method (MPGD-LDM) and two baselines (FreeDoM and LGD-MC). The user study consists of three parts: assessing the style consistency between the generated image and the reference style image, evaluating how well the generated image follows the text prompt, and the overall user preference.

We perform each part of this study with a separate questionnaire posted on Amazon Mechanical Turk. Each HIT task contains one multiple choice question. Example surveys are provided in 22. We generate 100 images from each method using randomly sampled WikiArt-PartiPrompts image-caption pair we create from the style guidance experiment. Each annotator is compensated with $0.12 USD for each HIT task, and we estimate the annotators to complete each HIT in 30 seconds to 1 minute, which yields an hourly earning rate of $7.2 to $14.4 USD.

The results are shown in the Table 5. Users' response regarding Style and Text generally align with the style score and CLIP score. Our method outperformed both methods in terms of overall user preference, which suggests that our method finds a better sweet spot balancing the text prompt condition and the style guidance.

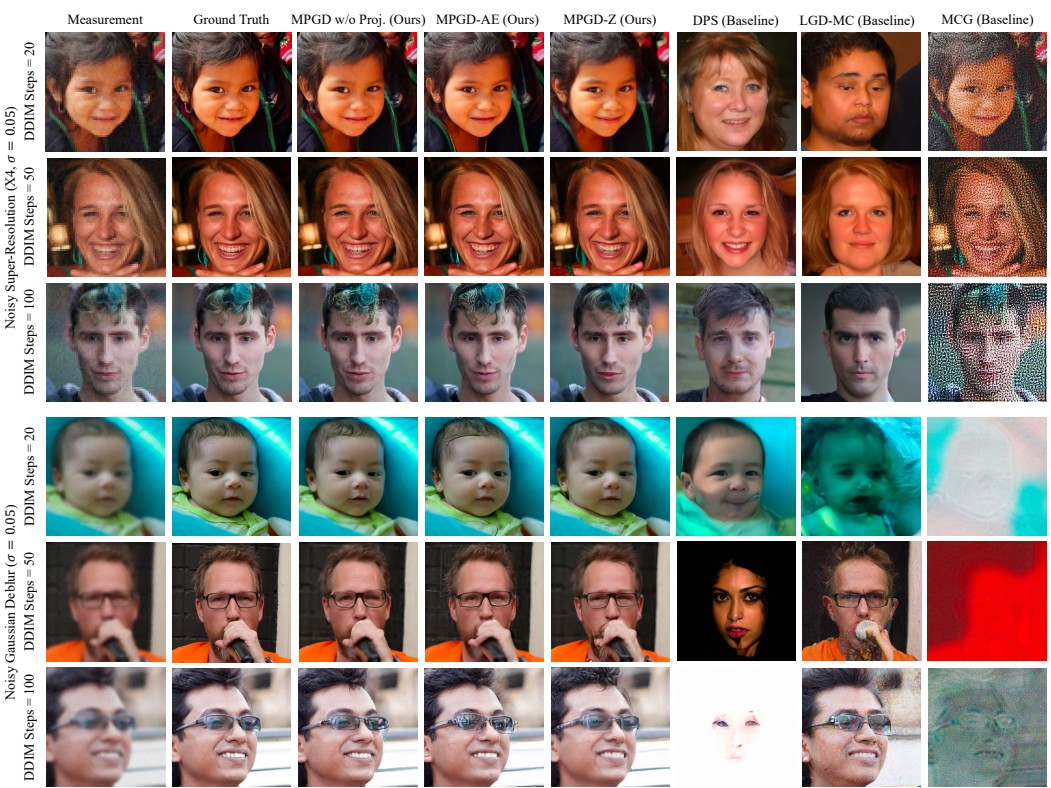

Figure 14: Additional qualitative examples of solving noisy linear inverse problems on FFHQ dataset.

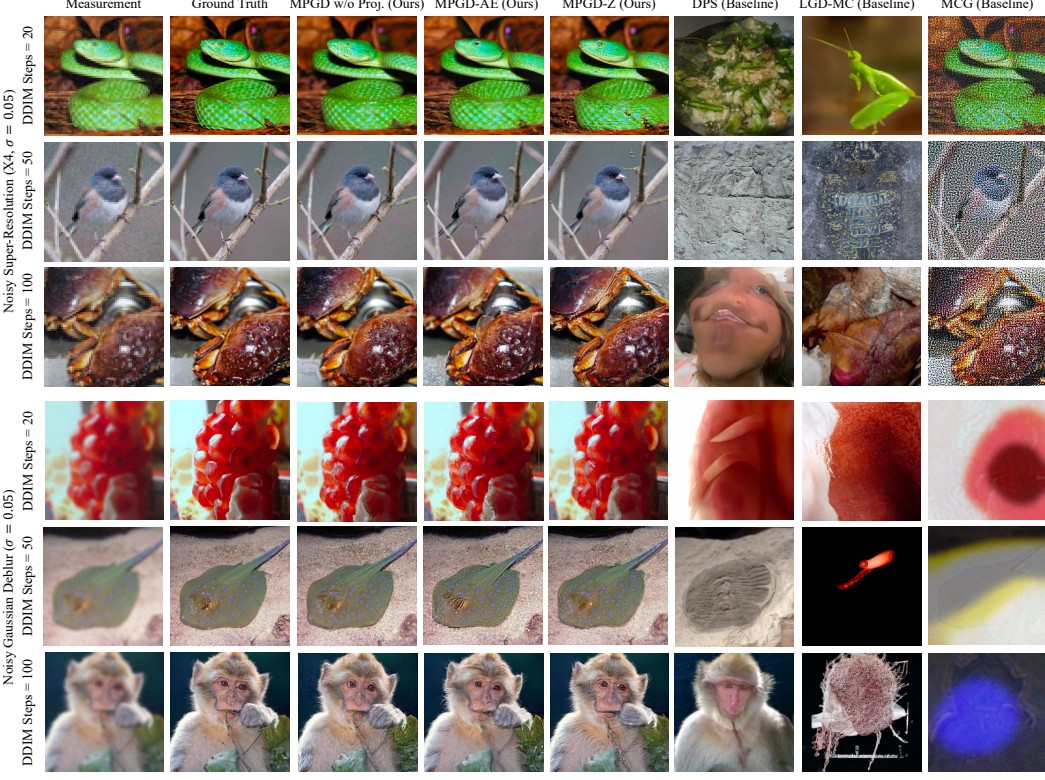

Figure 15: Additional qualitative examples of solving noisy linear inverse problems on ImageNet dataset.

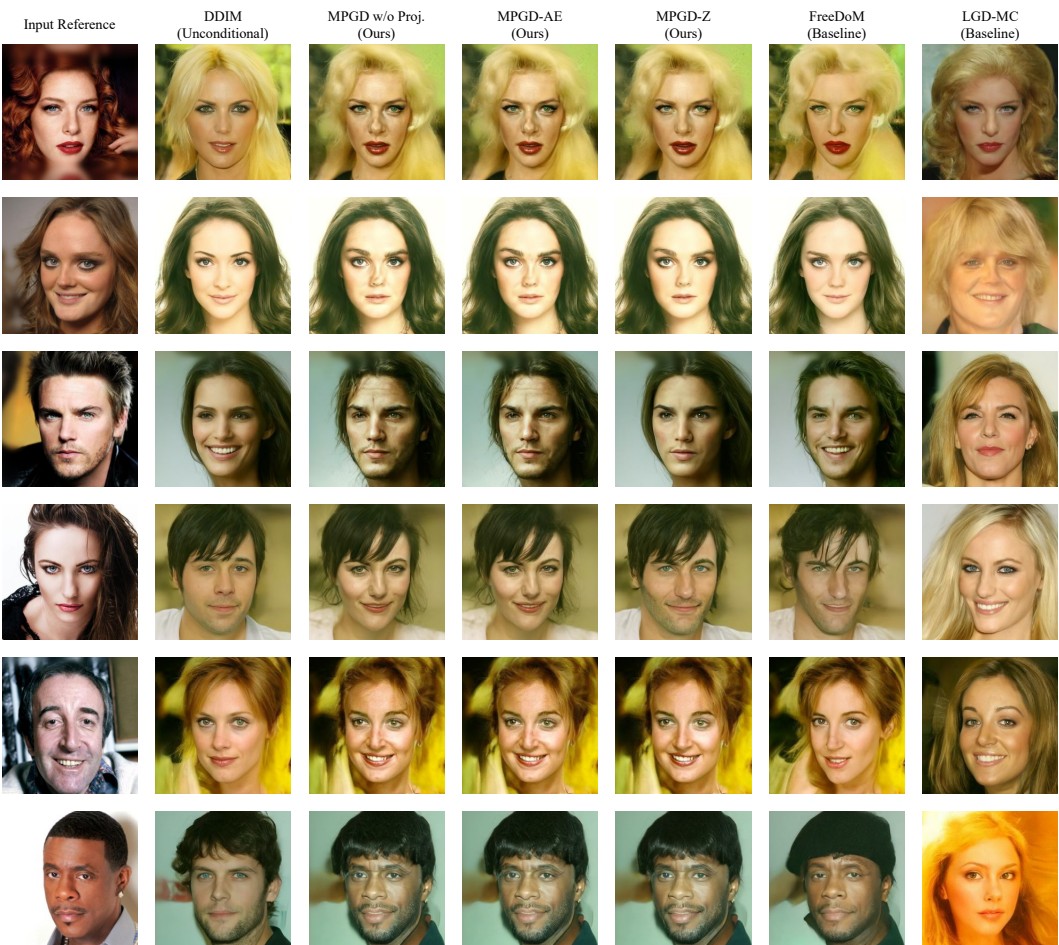

Figure 16: Additional qualitative examples of faceID guidance experiment.

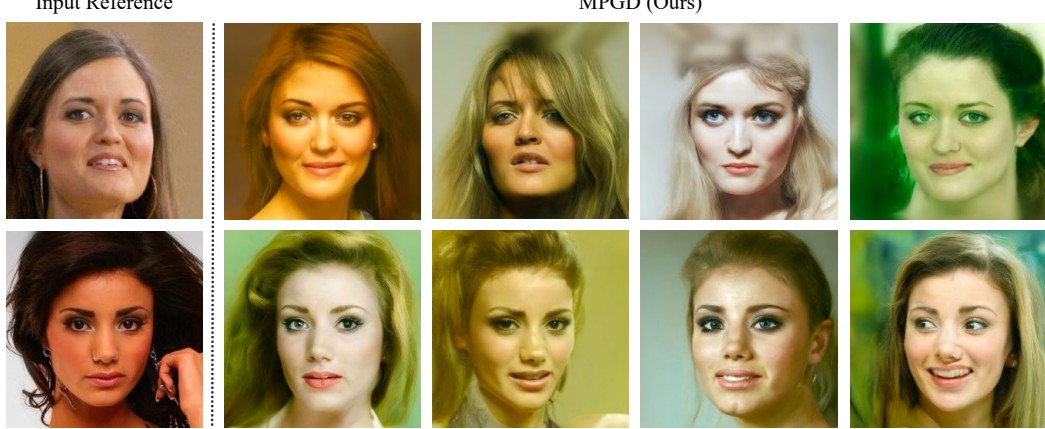

Figure 17: Additional qualitative examples of faceID guidance experiment to showcase the diversity of our generated images. With the same input reference image, our method is able to generate human face images that consist of the same identity as the reference image and that are diverse in many aspects including color, style and facial expressions.

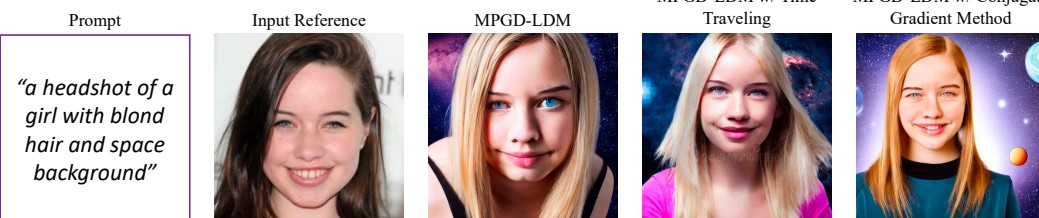

Figure 18: Results of FaceID guidance generation with Stable Diffusion and different optimization algorithms. In particular, non-linear conjugate gradient method improves the guidance quality while maintaining the fidelity, suggesting a promising direction for future investigation.

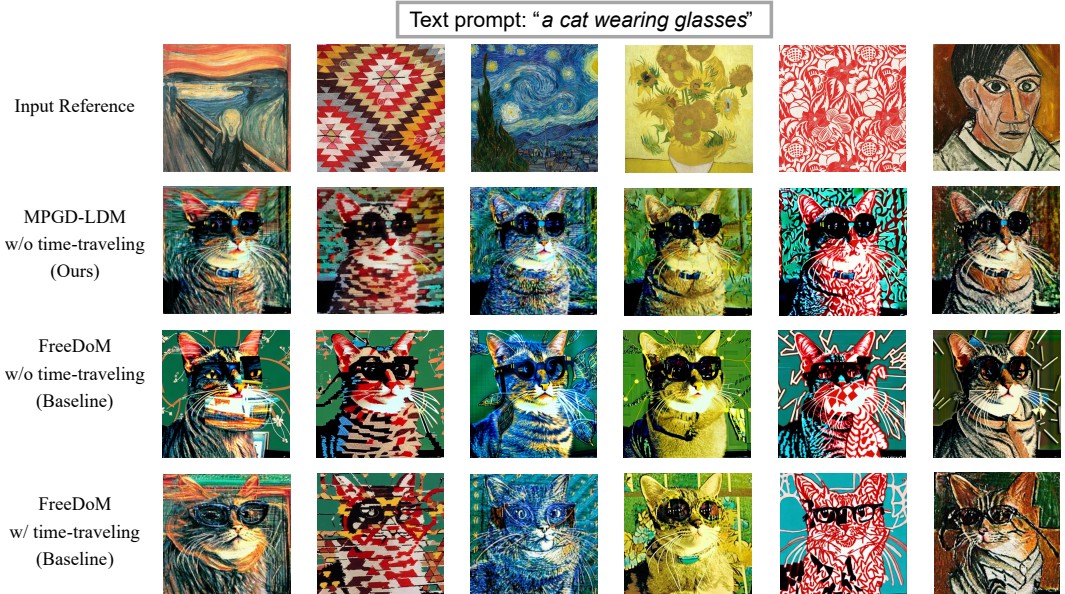

Figure 19: Additional qualitative examples of style guidance Stable Diffusion generation.

## F LIMITATIONS

### F.1 FAILURE CASES OF PIXEL-SPACE DIFFUSION GUIDANCE

In this section, we investigate the limitations and the failure modes of our proposed methods.

We observe common failures in solving noisy linear inverse problems with small numbers of DDIM steps. When the background of the image is predominantly white, prominent Gaussian noise-like patterns remain in the final results. Figure 23 shows an example of this kind of failure. While manifold projection can help mitigate the problem, we find that with small number of DDIM steps, it is usually not enough to completely reduce the Gaussian noise.

We also discover that the quality of the guided generation heavily depends on the performance of the guidance loss. If the loss function is not properly chosen, then it is very difficult to obtain satifactory results. For example, the ArcFace model is trained on centered and cropped humean headshot images that only recognizes the identity of a person by their facial landmark features. As a result, it is very hard to guide the sample to have other features such as skin tones that the model doesn't detect. However, these features are crucial for identifying an individual as well. Therefore, we advise our users to cautiously select the guidance loss functions.

### F.2 FAILURE CASES OF STYLE GUIDANCE GENERATION EXPERIMENT

In Figure 24, we present some notable failure cases encountered during the style guidance generation experiments with Stable Diffusion.

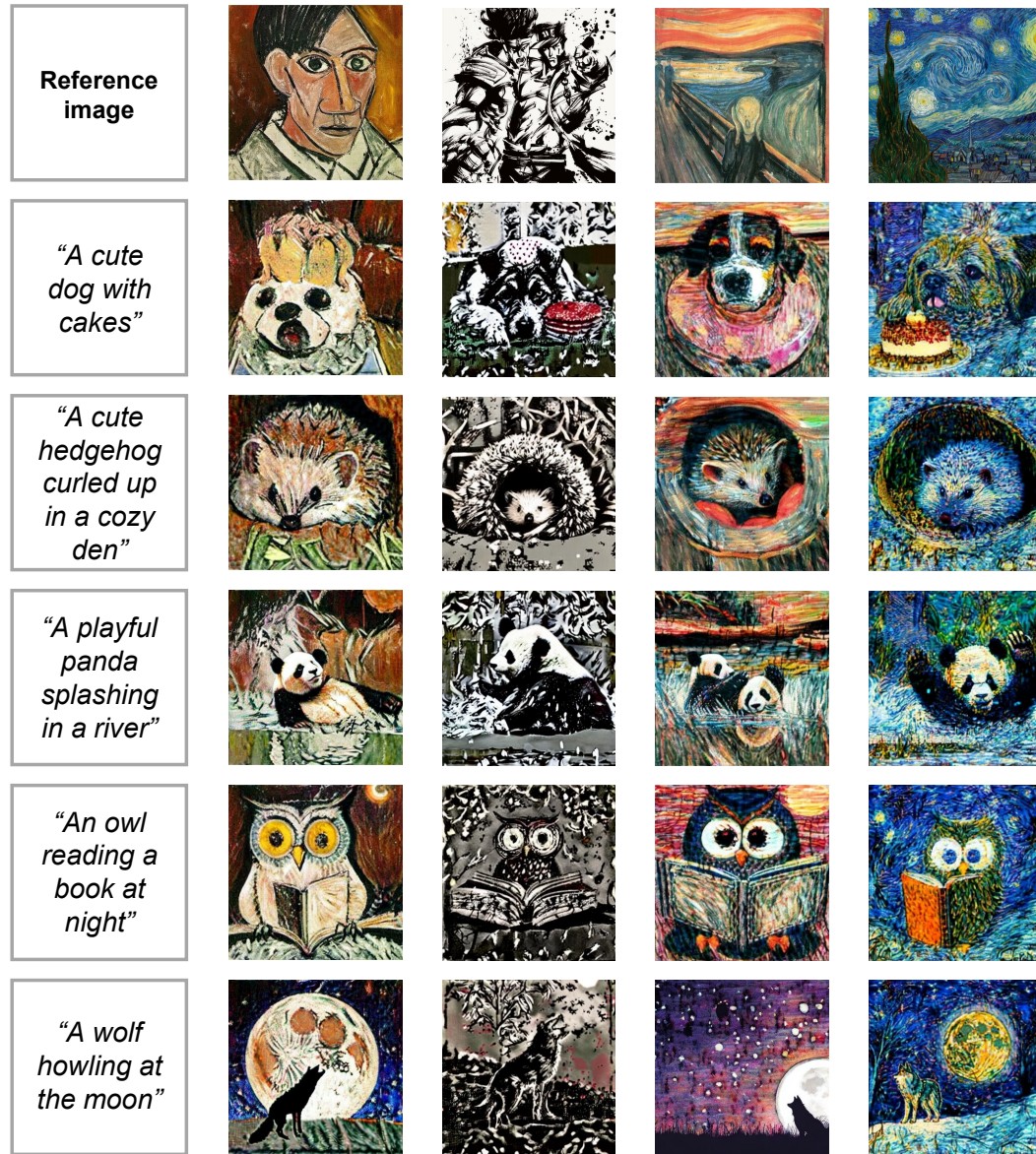

Figure 20: Additional qualitative examples of style guidance Stable Diffusion generation.

**1. Photo-realistic Outcomes from Painting References. (Fig. 24(a))** In this instance, despite the reference image being a realistic painting, the generated image resembles a photograph. This may be attributed to the inability of the loss function, used in this case, to effectively differentiate between a realistic painting and an actual photograph.

**2. Inadequate Reflection of Simplistic Reference Styles . (Fig. 24(b))** In this example, the reference image is a monochrome line drawing. However, the generated image, while partially capturing the color scheme, fail to replicate the style of the reference.

**3. Complex Prompts Leading to Incomplete Representation. (Fig. 24(c))** The prompt in thie example is "a corgi's head depicted as an explosion of a nebula." Without a style guide (i.e., in simple text-to-image generation), the "explosion of a nebula" aspect is evident in the generated image. However, in the result with the style guide, the aspect is not represented, likely due to a lack of correlation between the specified aspect from the prompt and the provided style.

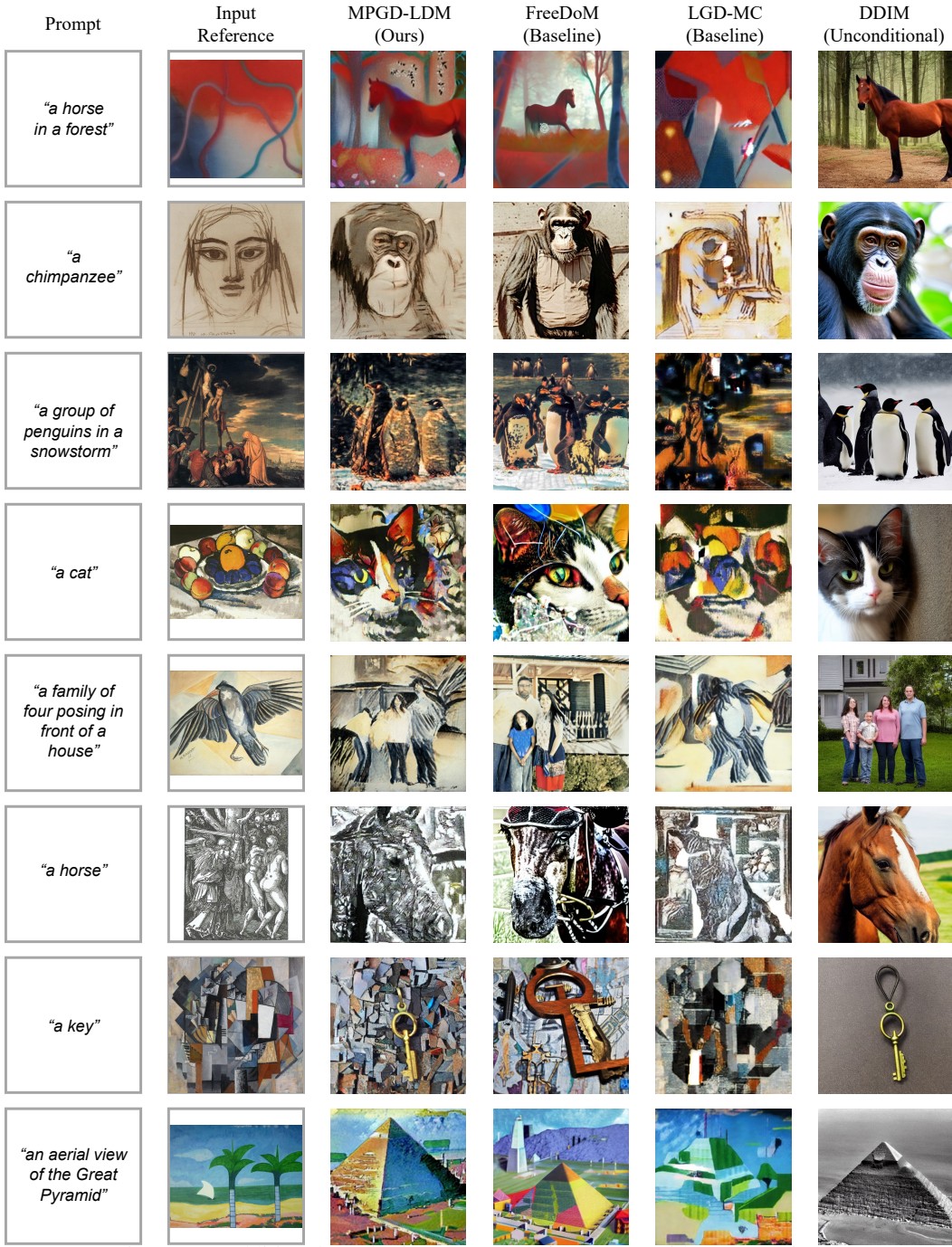

Figure 21: Additional qualitative examples of the style guidance experiment.

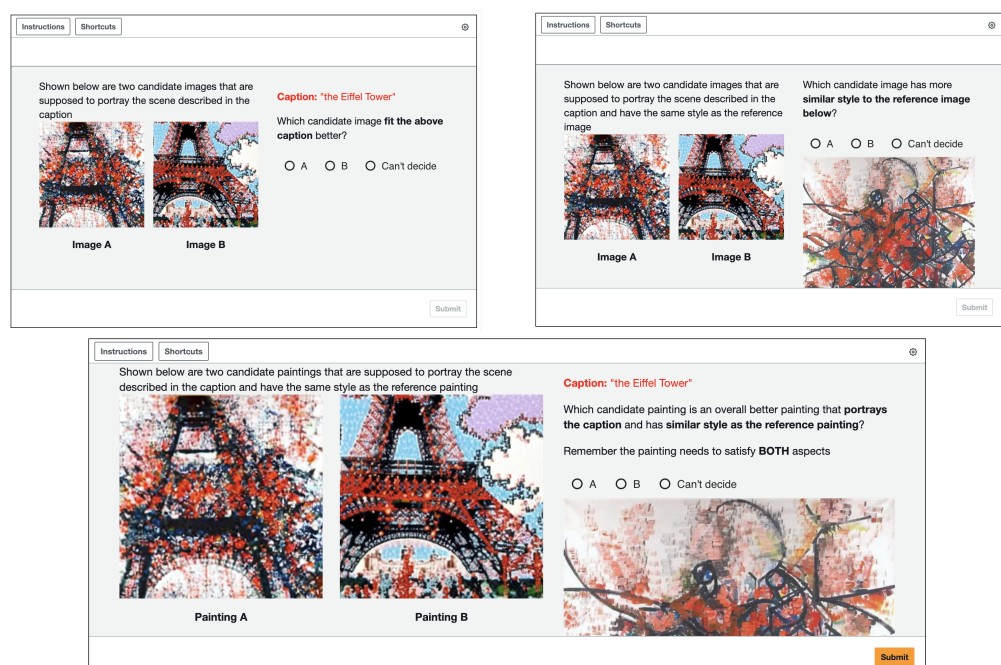

Figure 22: Example questionnaires we use in the user study.

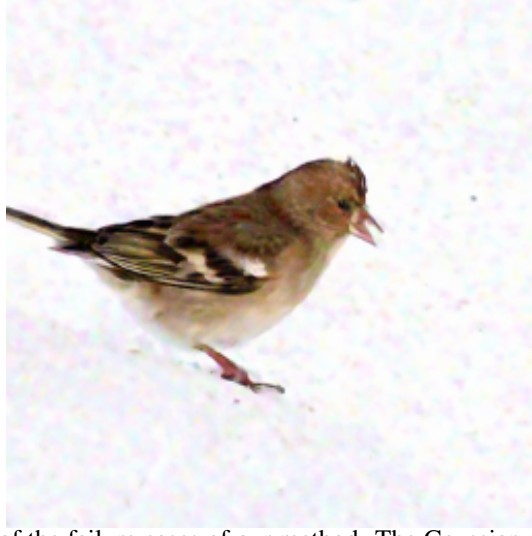

Figure 23: An example of the failure cases of our method. The Gaussian noises from the measurement remains in our reconstruction.

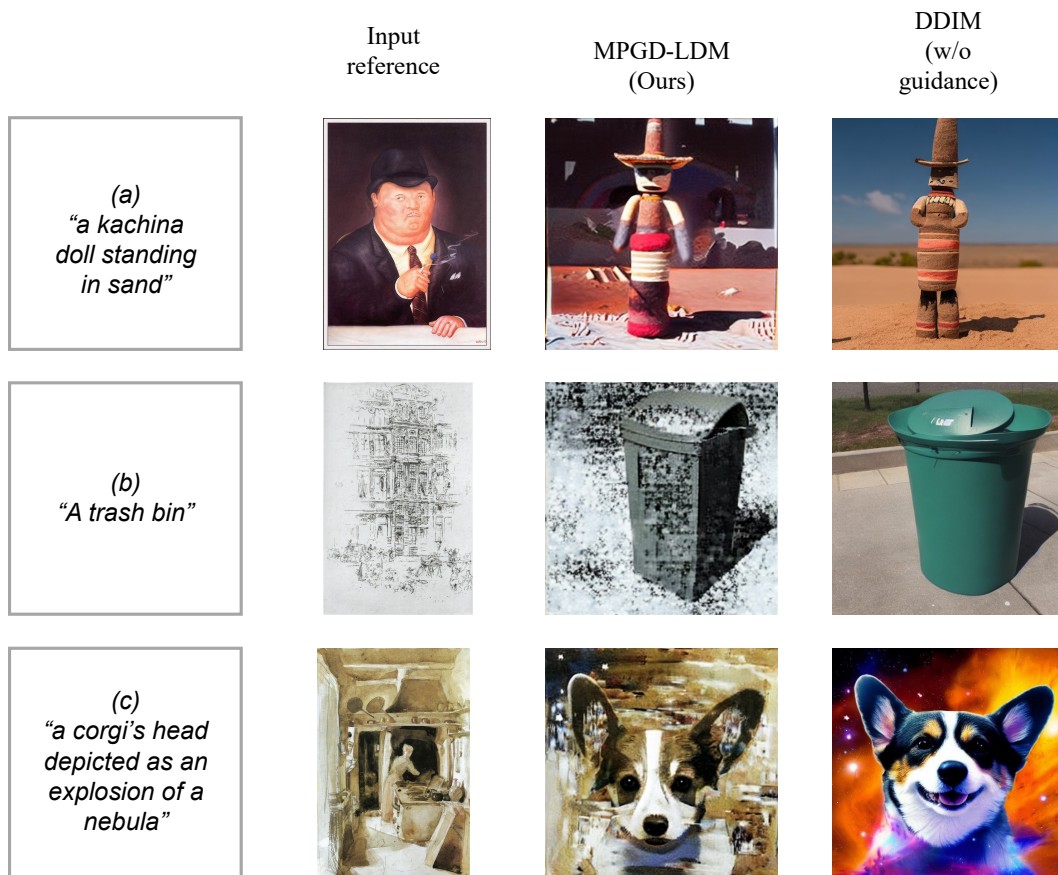

Figure 24: Noteworthy failure cases from images generated in the style guidance generation task with Stable Diffusion.

