# OpenReview forum: "Manifold Preserving Guided Diffusion"
_ICLR.cc/2024/Conference — ICLR 2024 poster_

### Official Review · Reviewer_smw4 · 2023-10-30

**Soundness:** 3 good
**Presentation:** 3 good
**Contribution:** 3 good
**Rating:** 6
**Confidence:** 3

**Summary:**

The paper introduces Manifold Preserving Guided Diffusion(MPGD), a framework that leverages the manifold hypothesis to enhance conditional generation tasks. By guiding the diffusion process using the underlying manifold structure, it achieves better sample quality and faster inference. The authors propose novel training-free methods applicable to both pixel and latent space models, demonstrating improved sample quality and inference speed.

**Strengths:**

The problem that traditional training-free guidance fails to guarantee the interfered results located in the manifold is quite interesting. This seems to exist in a wide range of existing approaches. Beisides, the proposed method is training-free, simple, efficient and easy to be implemented.

**Weaknesses:**

- Although the problem to be addressed looks quite fundamental and the presented approach is quite general, the experiments look quite limited. It only contains a restoration task, a face-id transferring task and a style guidance task, which is quite narrow considering that the proposed method is applicable to a lot of editing tasks. Besides, the paper repetitively uses several cases in both the manuscript and the appendix, which looks a bit boring.

- The results of style guidance look not that appealing.

- If possible, user study would make the experimental part stronger.

- There are some typos, e.g., in Eq. 2 (\alpha_t --> \alpha_{t-1}) and Fig. 14 (bitterfly --> butterfly).

**Questions:**

Please address my concerns or correct me if there is anything wrong in the weakness section.

---

> ### Author Response · Authors · 2023-11-17
> **Rebuttal Response**
>
> Your feedback for our work is greatly appreciated. In the following, we will address your questions and concerns in detail.
>
> 1. **Broader Applications and Repetitive Qualitative Showcase (Weakness 1):** Thank you for your suggestion. To further demonstrate the applicability of our method, we have added another experiment where we use a pre-trained CLIP model and text prompt to guide human face generation with pixel-space CelebA-HQ model. In Section E.1 and Figure 9 in the appendix, we show examples of samples guided by the text prompt and our method, in comparison with unconditional DDIM samples of the same random seed. Our method is able to create images that follow the text description provided while maintaining high fidelity. We have also updated our manuscript to include more qualitative examples for all experiments.
>
> 2. **Better Results for Style Guidance Stable Diffusion Generation (Weakness 2):** Based on Reviewer Yt4g’s suggestion, we were able to further improve the performance of our method on style guidance Stable Diffusion generation, and we have included new qualitative showcases in Figures 20 and 21 and quantitative results in Tables 2 and 4 in our paper.
>
> 3. **User Study (Weakness 3):** Thank you for your suggestion. We have conducted a user study for the style-guided Stable Diffusion generation task on Amazon Mechanical Turk to compare our method (MPGD-LDM) and two baselines (FreeDoM and LGD-MC). Details about the experiment are described in the appendix.
>
>     The results are shown in Table 5 in the appendix as well as below. Users’ responses regarding Style and Text generally align with the style score and CLIP score in Table 2. In terms of overall user preference, our method outperformed both methods, which suggests that our method finds a better sweet spot balancing the text prompt condition and the style guidance.
>
> |         Method        | Style Consistency (Win/Lose/Draw) | Prompt Consistency (Win/Lose/Draw) | Overall (Win/Lose/Draw) |
> |:---------------------:|:---------------------------------:|:----------------------------------:|:-----------------------:|
> | MPGD-LDM v.s. FreeDoM |             47%/45%/8%            |             32%/66%/2%             |        49%/45%/6%       |
> |  MPGD-LDM v.s. LGD-MC |             27%/64%/9%            |             69%/29%/2%             |        53%/43%/4%       |
>
> 4. **Typos (Weakness 4):** Thank you for pointing out the typos. We have corrected these errors in the revised manuscript.

---

> ### Author Response · Authors · 2023-11-21
> **Looking forward to your response**
>
> Thank you for your thorough and insightful review. As the rebuttal period is coming to its end, we are reaching out to see if you have any additional questions or concerns that we can address. Thank you and looking forward to your response.

---

> > ### Comment · Reviewer_smw4 · 2023-11-23
> >
> > Thanks for the authors' response and I apologize for the late reply. The response has addressed my concerns. Basically, I like the idea and the problem that the paper intends to overcome very much. And thus I will improve my score.

---

> > > ### Author Response · Authors · 2023-11-23
> > > **Thank you for your recognition of our work**
> > >
> > > Thank you for your response. We really appreciate your constructive feedbacks and your recognition of our work.

---

### Official Review · Reviewer_mY9K · 2023-10-31

**Soundness:** 3 good
**Presentation:** 3 good
**Contribution:** 3 good
**Rating:** 6
**Confidence:** 3

**Summary:**

This paper proposed a guided diffusion model based on manifold preserving. The basic idea is to insert the manifold constraint into the pre-trained diffusion model, by gradient-descent projection of the estimated clean image onto the manifold tangent space. The manifold constraint is implemented by the gradient descent of auto-encoder w.r.t. the estimated clean image. The paper presented theorems for the manifold preserving. The experiments mainly show that the proposed approach achieves high-quality generated image while having 3.8 times speed-up.

**Strengths:**

(1) The idea of imposing data manifold on the diffusion model using auto-encoder is an interesting idea.

(2) The paper presented sufficient proof for the manifold constraints.

(3) The generated images are overall good, especially with higher generation speed.

**Weaknesses:**

1. The paper claimed that the proposed MPGD can generate images with 3.8 times speed up compared with the baseline. The main paper should give more details on why it can speed up with the same number of iterations?

2.  The theoretical analysis on the auto-encoder in theorem 2 says that the gradient of auto-encoder lies on the tangent space of manifold. The analysis relies on the linear manifold hypothesis. Is this assumption realistic for real image set, and how to understand that the image manifold is linear?

3.  In the experiments for super-resolution, there are several methods using diffusion model for image super-resolution or other restoration methods, e.g., Denoising diffusion null-space model, etc. How about the comparisons with these Sota diffusion-based image inverse models?

4. The paper mentioned multi-step optimizations in the manifold projection. How does the number of iterations affect the conditional generation results in quality measures?

**Questions:**

Please refer to the weakness for my questions, mainly on the insufficient experimental justifications.

---

> ### Author Response · Authors · 2023-11-17
> **Rebuttal Response 1**
>
> We appreciate your valuable insights and acknowledgement of our efforts. Below, we respond to your questions and discuss the issues you have raised.
>
> 1. **Reason for Speedup (Weakness 1):** The reason for the speed-up is that our proposed MPGD does not require backpropagation through the diffusion model, which is generally known to be time-consuming. To be more specific, in DPS based methods, the guidance is calculated as $\nabla_{x_t} L(x_{0|t};y) = \nabla_{x_{0|t}} L(x_{0|t};y) \frac{\partial x_{0|t}}{\partial x_t}$ where $\frac{\partial x_{0|t}}{\partial x_t}$ requires backpropagation through the diffusion model. On the other hand, in MPGD, we only calculate the gradient $\nabla_{x_{0|t}} L(x_{0|t};y)$ with respect to $x_{0|t}$. Althoughin MPGD-AE and MPGD-Z, evaluating the autoencoder incurs additional costs, it is still faster than computing the gradients for the diffusion model itself. We have briefly discussed this reason in Section 4.2.1, and modified the writings in the experiment discussions to emphasize this reason.
>
> 2. **Linear Manifold Hypothesis (Weakness 2):** In response to your question about the linear manifold hypothesis in our theorem 2 analysis, it's important to clarify that our analysis of "the gradient of the auto-encoder lying on the tangent space of the manifold" does not actually depend on the linear manifold hypothesis. Proposition 1, which was initially proposed by [Chung et al. 2022], is based on this hypothesis.
>
>     While we acknowledge that the linear manifold hypothesis may not be entirely realistic for all complex real image datasets, it is often observed that this assumption holds locally. A practical example of this can be seen in image interpolation. Consider interpolating between two headshots of the same person, where one image features black hair and the other blond hair. The interpolation can result in another plausible headshot of the same person with brown hair. This example illustrates that even though the overall image manifold might not be linear, local linearity can be a reasonable approximation for certain kinds of image transformations.
>
>     Many prior works in the literature also make similar assumptions and implications. For example, the classic facial recognition paper [Turk et al., 1991] has discussed the validity of linear manifold hypothesis for specific subsets of image data such as well-aligned human face images. Hence, we believe this assumption can be made for the purposes of our analysis and is also meaningful for practical applications of our proposed methods.
>
> [Chung et al., 2022] Improving diffusion models for inverse problems using manifold constraints. NeurIPS 2022.
>
> [Turk et al., 1991] “Eigenfaces for Recognition”, Journal of cognitive neuroscience.

---

> ### Author Response · Authors · 2023-11-17
> **Rebuttal Response 2**
>
> 3. **Comparison with DDNM (Weakness 3):** Thank you for your suggestion. We have added the comparison experiment with DDNM on the FFHQ super-resolution task in Section E.2 in our paper. But before we dive into the discussion about the experiment, we would like to emphasize that DDNM is designed for only solving linear inverse problems, and it requires direct access to the operation matrix, its pseudo-inverse/SVD and the noise scale for the measurement, which are not parts of the assumptions that we have in our problem setting. As a result, DDNM is not applicable to the general setting of paper. Nevertheless, we also think it is valuable to better position our paper in the literature of linear inverse problem solving, and therefore we conducted the experiments described below.
>
>     To make a fair comparison, we use the simplified version of DDNM with no time traveling and the same unconditional diffusion model pre-trained by the authors of DPS, which is a smaller model compared to the one DDNM used in their paper. Due to code availability, we use average pooling as the interpolation method which is different from our original experiment setting. Figure 11 demonstrates the qualitative results, Figure 12 shows the enlarged details and Figure 10 shows the quantitative results. As we can observe in the figures, the simplified version of DDNM can achieve an equally fast sampling speed compared to our method and can also obtain similar guidance quality. However, the images generated from DDNM exhibit various artifacts, such as high-frequency circular patterns and overly smoothed generation. These artifacts prevent DDNM from maintaining high fidelity while our method can generate more realistic details.
>
>     That being said, we would like to point out that the generated images from DDNM tend to maintain better shapes whereas our method hallucinates small details more than DDNM. Therefore, when we compare PSNR values (Table 3), we can observe that DDNM is significantly better than our method. We also acknowledge that in the original paper of DDNM, in order to solve the noisy inverse problems, the authors suggest using multi-step time traveling to improve the performance which we did not deploy in order to make a fair comparison in terms of run time. Therefore, we believe that DDNM still has certain advantages over our method in terms of solving specific inverse problems. We think incorporating the consistency constraint in DDNM with our method can potentially achieve the best of both worlds, and will be an interesting future work direction.
>
> 4. **Multi-step Optimization (Weakness 4):** The effect of different numbers of optimization is indeed an interesting topic, and we found that the answer to this question actually depends on the tasks to which the users apply our method.
> Let’s assume we use time traveling as our optimization algorithm. Then, we can view it as a way to perform more Langevin steps in one diffusion step. Therefore, it has a higher chance of finding the optima, especially when the optima is further away from the original point. For example, suppose we want to sample “a headshot of a man” with CLIP guidance and pixel space CelebA-HQ model; then, if our unconditional DDIM sample is a headshot of a woman, then performing a multi-step optimization is going to be more beneficial since the samples that fit the prompt will be further away from the original unconditional sample. On the other hand, tasks like generating “a headshot of a person wearing red lipstick” only require small changes, and therefore, can be achieved in higher quality with fewer steps (see Figure 13). Generally speaking, the more challenging the tasks are (i.e. the more deviation required to get to the optima in expectation), the more beneficial multi-step optimization will be.
>
>     That being said, we do observe in practice that a large number of steps does not always benefit the generation. For example, we can start to observe unnatural artifacts appearing in the background of the image when sampling with 7 and 15 steps in the “red lipstick” experiment. In fact, we hypothesize that with a step size that is not infinitesimally small, asymptotically infinite-step optimization may lead to significant deviation from the data distribution. In addition, we can use different optimization algorithms such as nonlinear conjugate gradient and/or apply multi-step optimization to a selective subset of steps, following FreeDoM, and we also observe that the step sizes need to be adjusted accordingly for different numbers of steps. The asymptotic behavior of the multi-step optimization and how to select these hyper-parameters will be an interesting future work. We have included the discussion we have in this response in Section E.3 of our paper as well.

---

> ### Author Response · Authors · 2023-11-21
> **Looking forward to your response**
>
> Thank you for your thorough and insightful review. As the rebuttal period is coming to its end, we are reaching out to see if you have any additional questions or concerns that we can address. Thank you and looking forward to your response.

---

### Official Review · Reviewer_obSn · 2023-11-01

**Soundness:** 3 good
**Presentation:** 3 good
**Contribution:** 3 good
**Rating:** 6
**Confidence:** 4

**Summary:**

The paper introduces a novel way of generating conditional samples from unconditional diffusion models without additional fine-tuning. The method is based on modifying the DDIM sampling algorithm by performing gradient steps on the $x_0$ estimate with an off-the-shelf "clean-image" classifier.

**Strengths:**

- The authors propose a novel and flexible way to utilize the priors of large diffusion models in conditional generation tasks. The main advantage is the ability to use any off-the-shelf classifier to provide guidance, which previously relied on expensive, slow procedures that required multiple optimization steps during sampling.

**Weaknesses:**

- The experiments do not clearly demonstrate the idea of guiding the sampling process with an off-the-shelf network. In all three experiments, a reference image is used to guide the sampling. The projection to the manifold in these cases is shown to successfully guide the diffusion towards an image in the reference image neighborhood.  However, it is not clear how that would work in the case where the guidance is given by a more general differentiable loss $L$. This raises the question of whether the proposed method is applicable in cases other than the limited set of inverse problems presented, where the guidance may be less informative. An example would be an $L$ that measures the log-likelihood of an attribute in the image (e.g. blonde hair).

- The qualitative results in Figure 6 raise questions about the diversity of the samples drawn. The generated images shown, are almost all identical and deviate from the distribution of images in the original CelebA dataset (i.e. the colors are completely unnatural). The baseline also performs badly but the distribution seems closer to the original $p(x)$ that the model learned.

- The related works section should be in the main paper. This is necessary to draw clear distinctions between the existing works and the proposed approach and highlight what the authors' main contributions are.

**Questions:**

- Could the authors clarify what the $L$ function that is used in the different experiments is?

- Can the authors showcase their method on an experiment that is not based on sampling with a reference image? Is this a limitation of their method or can it be generalized to other formulations of $L$?

---

> ### Author Response · Authors · 2023-11-17
> **Rebuttal Response**
>
> Thank you for your constructive feedback. We would like to address your questions and concerns below.
> 1. **More General Guidance Loss w/o Reference Images (Weakness 1 & Question 2):** Thank you for bringing up this question. We would like to first clarify that our method is NOT limited to tasks with reference images, and therefore, we really appreciate you asking this question because although two of our original choices of tasks (FaceID guidance and style guidance generation) are not inverse problems, your question helped us realize that the previous example tasks we chose happen to all involve reference images. As a result, we have added an additional experiment where we use the CLIP model and text prompt to guide the pixel-space CelebA-HQ model. In Section E.1 and Figure 9 in the appendix, we show examples of samples guided by the text prompt and our method in comparison with unconditional DDIM samples of the same random seed. Our method is able to create images that follow the text description provided while maintaining high fidelity using an unconditionally pre-trained diffusion model.
>
> 2. **Diversity of the Generated Images (Weakness 2):** We apologize for the confusion caused by the presentation of this figure. Firstly, we would like to respectfully point out that the DDIM samples are the ones that showcase “the original p(x) that the model learned”. Therefore, in Figure 6 we can observe that our method is actually closer to the DDIM samples (and therefore the original p(x) that the model learned) regarding the color, structure, and other fine-grained details in the images. In fact, this is precisely because our method performs on the tangent spaces of the original DDIM clean data estimations, and therefore, is able to stay close to the original distribution while performing guided generation. We have also confirmed these qualitative observations with quantitative analysis in Table 1. The reason why these images seem to have strange color schemes is because the DDIM trajectories happen to produce these samples. We can also observe that FreeDoM, which is one of our baselines, generates similar images.
>
>     That being said, to demonstrate our method’s generation diversity, we have included Figure 16 and an additional Figure 17 in the appendix for the pixel space FaceID guidance generation experiment. In these figures, we demonstrate that images generated by our method show diversity in various aspects, such as color, style, and facial expression.
>
> 3. **Related Works (Weakness 3):** Thank you for your feedback. We initially structured our paper to address the most relevant prior works and their limitations in Sections 2 and 3 before introducing our method in order to adhere to the page constraints while reserving an in-depth exploration of further related literature for the appendix. Based on your suggestion, we have now added Section 2.4 in our main paper to briefly discuss more related works while keeping the detailed discussion in the appendix.
>
> 4. **Specifications of loss functions used in experiments (Question 1):** We apologize for the confusion. Regarding the style-guided generation task, we recognized that our definition of the loss function was not as clear as it should have been. To rectify this, we have provided additional details in Appendix D for further clarification. To summarize, the settings for the super-resolution task are the same as those used in DPS [Chung et al., 2023], and the settings for the FaceID guidance generation and text-to-image style guidance generation are the same as those employed in FreeDoM [Yu et al., 2023].
>
>
> [Chung et al., 2023] “Diffusion posterior sampling for general noisy inverse problems”, In Proc. ICLR, 2023.
>
> [Yu et al., 2023] “FreeDoM: Training-free energy-guided conditional diffusion model”, In Proc. CVPR, 2023.

---

> > ### Comment · Reviewer_obSn · 2023-11-20
> >
> > I would like to thank the authors for their thorough response.
> >
> > 1. **On generality:** I apologize for misleading with the *inverse problems* term, as it usually refers to linear inverse problems. My concern was regarding the nature of the experiments, which was limited to using a reference image. The authors have addressed this concern and I would strongly encourage them to include their CLIP-guided generation examples in the main text as it is an important addition and can help with the overall understanding of the capabilities of the method.
> >
> > With all of the weaknesses addressed, I will be increasing my score.

---

> > > ### Author Response · Authors · 2023-11-21
> > > **Thank you for your recognition of our work**
> > >
> > > Thank you for your recognition of our work, and thank you for the feedbacks! We have updated our paper so that our introduction figure (Figure 1) includes an example from the CLIP guidance generation experiment. We really appreciate your suggestions in your review and your updated score.

---

### Official Review · Reviewer_Yt4g · 2023-11-01

**Soundness:** 4 excellent
**Presentation:** 4 excellent
**Contribution:** 3 good
**Rating:** 8
**Confidence:** 3

**Summary:**

The paper applies the manifold hypothesis to enhance the control in unconditional diffusion models. One key finding from the paper is that the guidance term in the diffusion process can potentially have a detrimental effect on the evaluation of the score function. In other words, it can cause deviations from the data manifold and lead to the generation of unrealistic images. To address this issue, the paper suggests a solution where the guidance gradient of clean samples is projected onto the tangent space of the manifold using an autoencoder, allowing the model to stay closer to the manifold.

**Strengths:**

- The paper presents a versatile idea that can be integrated with existing methods. This approach offers a balance between efficiency and quality, showcasing lower overhead compared to baselines, while also demonstrating improved quality and faster inference speed.
- The paper provides valuable insights, and these insights are effectively supported by experiments, as in Section 3.
- The development of the method in Section 4 is presented in a clear and accessible manner. The logical flow and explanations make it easy for readers to follow the methodology.
- The paper demonstrates thoroughness in its presentation of baselines for experiments in both the pixel space and latent space. The dataset choices are reasonable, coupled with compelling figures and metrics, enhances the persuasiveness of the results.

**Weaknesses:**

I don't see major weaknesses in the paper, it is cohesive and well-rounded.

- While the paper appropriately utilizes KID as a less biased metric, it could benefit from including FID (Fréchet Inception Distance) as a more widely recognized and standardized metric for related work comparison. The absence of FID makes it challenging for readers to assess the method's performance in relation to existing or future approaches.
- The paper misses the opportunity to provide insight into failure cases when it comes to text-guided generation. This is particularly important given its widespread use in the community.

**Questions:**

- What is the method's guidance consistency in comparison to DDIM?
- Is there any behavior change in the classifier-free guidance scale for text in the style guided experiments?

---

> ### Author Response · Authors · 2023-11-17
> **Rebuttal Response**
>
> Thank you for your constructive feedback and your recognition of our work. We would like to answer your questions and address your concerns below.
> 1. **FID (Weakness 1):** We agree that it would be helpful to include FID since it is a more popular metric in the community. However, we would like to emphasize that for FID to be reliable, it would require at least 10k samples to evaluate for each experiment [Bińkowski et al., 2018]. Given the limited timeframe of the rebuttal period and the required sampling time for all the methods compared, it is not very feasible for us to produce a large enough number of samples to make proper FID comparisons, and we respectfully propose deferring this extensive analysis to future work, to ensure thorough and accurate evaluation.
>
> 2. **Additional Failure Cases (Weakness 2):** Based on your comment, we have added some noteworthy failure cases from the images generated in the style transfer task with StableDiffusion and have included a discussion in the Appendix F.2. In summary, there exist some instances where the expected style is not reflected in the generated images due to the design of the loss function or the relationship between the provided prompt and the reference style.
>
> 3. **Comparison with DDIM (Question 1):** We apologize for the confusion. In the context of our paper, DDIM performs unconditional sampling from the learned diffusion model whereas our method receives extra guidance from an external loss function provided by the users to perform conditional generation. In Figure 2, we illustrate the relationship between DDIM and our method, and we also include comparison experiments between DDIM and our method in the FaceID guidance generation and style guidance Stable Diffusion generation experiments. This comparison demonstrates how our method modifies the original unconditional DDIM sampling trajectory to generate conditional samples with respect to the loss guidance while keeping the samples on the correct manifolds, hence maintaining high sample quality.
>
> 4. **Effect of Classifier-Free Guidance Scale (Question 2):** Thank you for this question. In the appendix E.4 and the table below, we have included an additional quantitative comparison in style guided Stable Diffusion generation with various classifier-free guidance scales (CFG). This analysis actually helped us to produce better results with our method in this experiment, and we have also modified our main paper to include the updated results with further hyperparameter tuning. It's worth noting that the CFG scale has a positive impact on the style score, and strong CFG scales appear to help decrease the loss function. However, although in vanilla text-to-image generation tasks a larger CFG scale tends to lead to a higher CLIP score [Nichol et al., 2022], since the distribution we aim to sample from is also guided by another external loss function, we do not observe the same trend in our setting. In practice, we would suggest our users to adjust this hyperparameter to suit their preference of tradeoff between the style guidance and the text prompt condition.
>
>
> |MPGD-LDM |**Style**$\downarrow$|**CLIP**$\uparrow$|
> |----|----|----|
> |CFG=2.5|493.5|26.98|
> |CFG=5.0|459.8|**27.08**|
> |CFG=7.5|**441.0**|26.61|
>
>
> [Nichol et al., , 2022] “GLIDE: Towards Photorealistic Image Generation and Editing with Text-Guided Diffusion Models”, Proceedings of the ICML, 2022.

---

> ### Author Response · Authors · 2023-11-21
> **Looking forward to your response**
>
> Thank you for your thorough and insightful review. As the rebuttal period is coming to its end, we are reaching out to see if you have any additional questions or concerns that we can address. Thank you and looking forward to your response.

---

> ### Comment · Reviewer_Yt4g · 2023-11-22
>
> Thank you for your detailed response.
>
> - **Comparison with DDIM**: Apologies for stating Question 1 so poorly. I meant to ask about the consistency of guidance when using the proposed method versus DDIM. In any case, this can now be inferred by examining the added failure cases.
> - **Effect of Classifier-Free Guidance Scale**: I'm happy the analysis led to better results, and with the analysis itself.
>
> Although FID evaluations are still missing, I find my concerns addressed properly and I maintain my score.

---

### Author Response · Authors · 2023-11-17
**Message to the reviewers**

First of all, we would like to express our deep appreciation to all the reviewers who have dedicated their time and expertise to reviewing our manuscript. We've thought carefully about every comment and question you've all raised. You'll find our detailed responses to these points just below.

In response to your comments, we have made the following changes to our manuscript. Below is a summary of the modifications:
1. We corrected several typos and refined the writings in our theoretical statements and experiment discussions to improve readability.
2. We added a subsection on related works in the main text. Due to space constraints, details are still described in the appendix.
3. As an additional experiment, we conducted CLIP-guided generation, which we discuss in detail in the appendix.
4. We performed a comparison with Denoising Diffusion Null-space Models (DDNM), which is described in the appendix.
5. We investigated the impact of the number of optimization steps on the generation results, with findings noted in the appendix.
6. We examined the effect of the classifier-free guidance scale on text-to-image models in style guidance generation experiments, with details in the appendix.
7. We have added and updated several qualitative result showcases.
8. We conducted a user study on the style guidance generation task, with details included in the appendix.
9. We expanded the "Details on Experiments" section in the appendix.
10. We added a discussion on the failure cases of the style guidance generation task in the appendix.

---

### Meta-Review · Area_Chair_ADmZ · 2023-12-08

**Metareview:**

This research introduces a novel conditional image generation framework known as Manifold Preserving Guided Diffusion (MPGD). The framework is unique in that it requires no additional training and utilizes pre-existing diffusion models and readily available neural networks. This results in minimal additional inference costs, making it suitable for a wide variety of tasks. The researchers have drawn upon the manifold hypothesis to refine the guided diffusion steps, introducing a shortcut algorithm in the process. Two methods are proposed for on-manifold, training-free guidance using pretrained autoencoders. The key solution lies in that the guidance gradient of clean samples is projected onto the tangent space of the manifold using an autoencoder, utilizing the manifold hypothesis. The experimental results show that MPGD can be used in low-compute settings for a variety of conditional generation applications, with speed-ups of up to 3.8 times while maintaining high sample quality compared to existing baseline methods. This indicates that MPGD could have broad applicability and offers a significant improvement in both speed and cost-effectiveness for conditional image generation tasks.

**Justification For Why Not Higher Score:**

One reviewer thinks the paper good, while the majority three think it above the acceptance threshold.

**Justification For Why Not Lower Score:**

The reviewers unanimously accepted the paper.

---

### Decision · Program_Chairs · 2024-01-16

Accept (poster)